# Rhs NADase effectors and their immunity proteins are exchangeable mediators of inter-bacterial competition in *Serratia*

Martin Hagan[1,2], Genady Pankov[1,2], Ramses Gallegos-Monterrosa[1], David J. Williams [1], Christopher Earl[1], Grant Buchanan[1], William N. Hunter [1] ✉ & Sarah J. Coulthurst [1] ✉

Many bacterial species use Type VI secretion systems (T6SSs) to deliver anti-bacterial effector proteins into neighbouring bacterial cells, representing an important mechanism of inter-bacterial competition. Specific immunity proteins protect bacteria from the toxic action of their own effectors, whilst orphan immunity proteins without a cognate effector may provide protection against incoming effectors from non-self competitors. T6SS-dependent Rhs effectors contain a variable C-terminal toxin domain (CT), with the cognate immunity protein encoded immediately downstream of the effector. Here, we demonstrate that Rhs1 effectors from two strains of *Serratia marcescens*, the model strain Db10 and clinical isolate SJC1036, possess distinct CTs which both display NAD(P)$^+$ glycohydrolase activity but belong to different sub-groups of NADase from each other and other T6SS-associated NADases. Comparative structural analysis identifies conserved functions required for NADase activity and reveals that unrelated NADase immunity proteins utilise a common mechanism of effector inhibition. By replicating a natural recombination event, we show successful functional exchange of CTs and demonstrate that Db10 encodes an orphan immunity protein which provides protection against T6SS-delivered SJC1036 NADase. Our findings highlight the flexible use of Rhs effectors and orphan immunity proteins during inter-strain competition and the repeated adoption of NADase toxins as weapons against bacterial cells.

Bacteria typically exist in polymicrobial communities where they compete with closely- and distantly related microbes for space and resources, often by directly killing or disabling competitor cells. A widespread and important weapon used for competition between Gram-negative bacteria is the Type VI secretion system (T6SS), a large, contractile nanomachine used to deliver toxic effector proteins into neighbouring cells[1]. The T6SS functions by propelling a rigid puncturing structure decorated with effector proteins out of the secreting cell and into an adjacent recipient cell. This puncturing structure is composed of a tube of stacked rings of Hcp proteins, topped with a

spike made from a VgrG trimer and a single PAAR (proline-alanine-arginine repeat) domain-containing protein. The expulsion of this structure from the secreting cell is driven by contraction of an extended cytoplasmic sheath-like structure anchored in a trans-membrane basal complex[2,3]. Effector proteins associate with components of the puncturing structure either through non-covalent interactions (cargo effectors) or by fusion of effector domains to core Hcp, VgrG or PAAR domains (specialised effectors)[2]. The T6SS can be used against eukaryotic cells, including host cells and fungal competitors, or to scavenge nutrients from the extracellular environment. However, the

[1]School of Life Sciences, University of Dundee, Dow Street, Dundee DD1 5EH, UK. [2]These authors contributed equally: Martin Hagan, Genady Pankov.
✉e-mail: w.n.hunter@dundee.ac.uk; s.j.coulthurst@dundee.ac.uk

primary role of the T6SS appears to be during inter-bacterial competition, where it delivers multiple anti-bacterial effector proteins into a neighbouring competitor, causing death or inhibition of growth[1,4]. Many T6SS-delivered anti-bacterial effector proteins have been described, including families of peptidoglycan amidase and glycoside hydrolase effectors, phospholipases and pore-forming effectors, DNA hydrolases and deaminases, and effectors interfering with cellular cofactors[2,5]. In order to prevent intoxication of self or genetically-identical neighbour cells, secreting cells possess specific immunity proteins cognate to each anti-bacterial effector and encoded by the adjacent gene. These immunity proteins reside in the compartment of action of their effector protein and neutralise toxicity by tight and specific binding to the effector[1].

The first T6SS effector reported to target an essential cellular cofactor was Tse6 from *Pseudomonas aeruginosa*, a PAAR-containing specialised effector whose C-terminal domain displays NAD(P)$^+$ glycohydrolase (NADase) activity[6]. A subsequent study identified another NAD(P)$^+$ glycohydrolase effector from *Pseudomonas protegens*, Tne2, and suggested that Tse6 and Tne2 represent founder members of two related families of NADase effectors, Tne1 and Tne2, respectively[7]. NADase toxins deplete available NAD(P)$^+$, an abundant cofactor essential for cellular function in all kingdoms of life. Bacterial NADase toxins are also used against host cells, exemplified by the *Mycobacterium tuberculosis* tuberculosis necrotising toxin (TNT), an NADase domain cleaved from the CpnT (channel protein with necrosis-inducing toxin) protein. TNT induces macrophage cell death and is part of a distinct family of NADases found in many bacterial and fungal pathogens[8].

The T6SS can be used for competition between closely related strains of the same species and there is considerable variation in effector-immunity genes within, as well as between, species[1,9]. Additionally, orphan immunity proteins lacking a cognate effector are believed to provide protection against effectors delivered by competitors[10], although experimental evidence to support this idea is currently limited. Inter-strain variation and plasticity in T6SS effectors is exemplified by the rearrangement hotspot (Rhs) class of specialised effectors. Rhs proteins are large, polymorphic toxins with a highly variable C-terminal toxin domain (CT) preceded by a conserved Rhs-repeat containing domain which forms a shell-like structure around the CT[11]. T6SS-associated Rhs proteins have an N-terminal region containing a PAAR domain and structures important for chaperone binding and target cell entry[12]. Rhs CTs have a range of predicted or demonstrated activities, including varied DNase and RNase domains. The cognate immunity protein (RhsI) is always encoded immediately downstream of the *rhs* gene, allowing CT-I units to be exchanged through homologous recombination in the conserved *rhs* regions[1]. Whilst Rhs proteins containing Tne2-family NADase CTs have been predicted bioinformatically[7], Rhs-associated NADase activity has not yet been demonstrated.

*Serratia marcescens* is an opportunistic bacterial pathogen which occupies diverse environmental niches and represents a significant cause of hospital-acquired infections[13]. The model strain *S. marcescens* Db10 has a single T6SS which displays anti-bacterial and anti-fungal activity and delivers at least ten effector proteins, including two Rhs proteins[14,15]. Previous work showed that Rhs1 is dependent on the VgrG2 spike protein and a specific chaperone, EagR1, for delivery and that its CT (Rhs1CT$_{Db10}$) is a cytoplasmic-acting anti-bacterial toxin[14,16]. However, the mode of action of Rhs1CT$_{Db10}$ has remained unclear. Rhs1CT$_{Db10}$ is neutralised by the immunity protein, RhsI1$_{Db10}$, encoded immediately downstream of *rhs1*, whilst the function of the proteins encoded by two other small genes immediately downstream of *rhsI1$_{Db10}$* at the 3′ end of the T6SS gene cluster is unknown. In this study, we use a combination of structural, biochemical and genetic approaches to show that Rhs1CT$_{Db10}$ is an NAD(P)$^+$ glycohydrolase toxin and is directly inhibited by RhsI1$_{Db10}$, with RhsI1$_{Db10}$ representing

a previously undescribed family of immunity proteins. Additionally, we report that a clinical isolate of *S. marcescens*, strain SJC1036, has an Rhs1 protein whose CT (Rhs1CT$_{1036}$) represents a distinct family of NAD(P)$^+$ glycohydrolases, emphasizing the broad utility of such toxins in bacterial competition. Finally, by engineering Rhs1 from *S. marcescens* Db10 to deliver Rhs1CT$_{1036}$, we demonstrate that one of the genes downstream of *rhsI1$_{Db10}$* in Db10 encodes an orphan immunity protein able to protect against Rhs1CT$_{1036}$ and provide evidence for a patchwork of in-use and orphan immunity proteins in the Rhs1 locus in *Serratia*.

## Results

### The crystallographic structure of the Rhs1CT$_{Db10}$-RhsI1$_{Db10}$ effector-immunity complex

In preliminary work, remote protein homology and structural prediction suggested a possible distant relationship between Rhs1CT$_{Db10}$ from *S. marcescens* Db10 and the CTs of Tse6 and Tne2, but there was insufficient similarity to assign function. Therefore, in order to elucidate the molecular function of this effector domain, we initiated a structural study. His$_6$-tagged Rhs1CT$_{Db10}$ (His$_6$-Rhs1CT$_{Db10}$, amino acids 1333-1473 of full length Rhs1) was co-expressed with its immunity protein, RhsI1$_{Db10}$, and a stable 37 kDa heterodimeric His$_6$-Rhs1CT$_{Db10}$-RhsI1$_{Db10}$ complex was isolated, demonstrating a direct physical interaction between the effector and immunity protein. The Rhs1CT$_{Db10}$-RhsI1$_{Db10}$ complex was purified to homogeneity, crystallised and the crystal structure determined to 1.3 Å resolution. The phases for the first electron density map were determined based on the resonance scattering of bromide ions that were soaked into the preformed crystals.

The Rhs1CT$_{Db10}$-RhsI1$_{Db10}$ complex is a globular entity with approximate dimensions of $50 \times 45 \times 60$ Å. The structure places Rhs1CT$_{Db10}$ as a member of the broad TNT-like family of bacterial NADases, which includes TNT, Tse6 and Tne2, displaying the palm domain characteristic of this enzyme family[8]. The palm domain is dominated by a complex, twisted β-sheet comprising seven strands in order 1-5-6-3-2-4-7, aligned in anti-parallel fashion (Fig. 1). There are six helices, the N-terminal α1, then α2 and α3 that link β3 with β4, then a short α4 is present between β5 and β6 creating a cavity, and finally α5 that precedes the C-terminal β7. The ~18 Å groove extending into the core of the protein along β5 and β6 represents the putative active site. This cavity is lined by residues on the turn between α1 and β1, strands β2 and β3 including the short turn linking these two elements of secondary structure, β6, and a loop between α3 and β4. The immunity protein RhsI1$_{Db10}$ displays an α/β fold not observed previously, although in similar fashion to the effector domain, it is founded around a seven stranded β-sheet. The strands are anti-parallel in the order 4-5-6-7-10-9-8 (Fig. 1). On one side of the β-sheet, on the surface of the effector-immunity complex, lies the N-terminal segment of the immunity protein which adopts a three stranded antiparallel sheet, in order 1-2-3, then a loop including a short α1 that leads to β4. The other side of the main sheet, together with contributions from the loops linking β7 with β8, β9 with β10, and extending from β10 to α4, forms the interface to interact with and block the activity of Rhs1CT$_{Db10}$ by occluding the putative catalytic cleft (Fig. 2a).

The protein:protein interface in the Rhs1CT$_{Db10}$-RhsI1$_{Db10}$ complex covers approximately 1440 Å$^2$ and uses 15% of the solvent accessible surface area of Rhs1CT$_{Db10}$ (9435 Å$^2$) and 20% of RhsI1$_{Db10}$ (7315 Å$^2$). These values are indicative of a stable protein-protein complex[17]. The number of residues involved in interactions with the partner are 37 (about 28% of total residues) and 43 (about 26.5% of total residues) from Rhs1CT$_{Db10}$ and RhsI1$_{Db10}$, respectively. These residues form 16 direct hydrogen bonds and 10 salt bridges between the partners in addition to extensive van der Waals interactions. There are also numerous water-bridged hydrogen bonding interactions that will contribute to the association of the partners (Supplementary

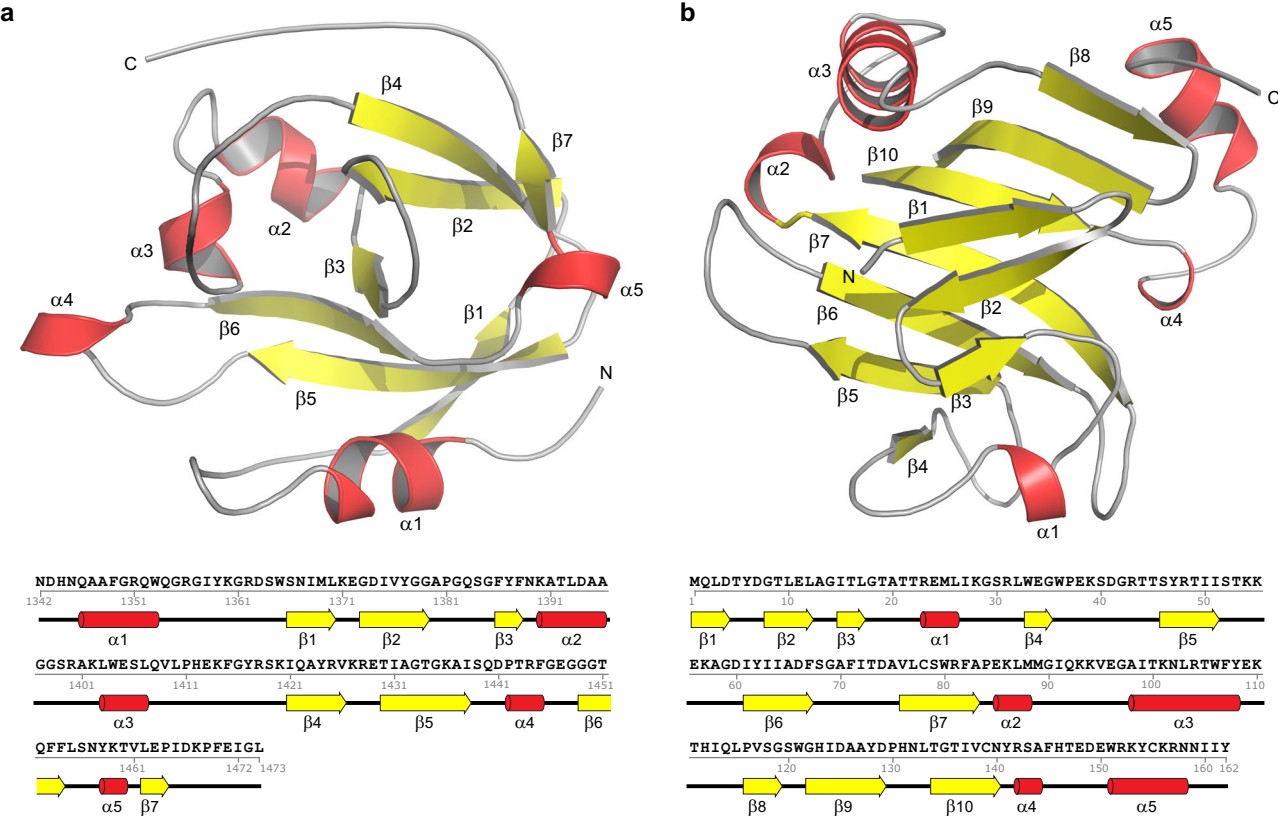

**Fig. 1 | The structures of Rhs1CT$_{Db10}$ and RhsI1$_{Db10}$.** Cartoon representation of the structures of (**a**) Rhs1CT$_{Db10}$ and (**b**) RhsI1$_{Db10}$. Helices are shown in red, strands in yellow and loops in grey. Below each structure is the amino acid sequence showing assigned elements of secondary structure.

Fig. 1). The calculated free energy of complex formation, ΔG, is −13.6 kcal/mol, a value consistent with other T6SS effector-immunity pairs such as Tne2$_{CT}$-Tni2 at −12.4 kcal/mol and Tse6$_{CT}$-Tsi6 at −9.6 kcal/mol[7] and these values are again consistent with the formation of a highly stable complex.

Of particular note is the RhsI1$_{Db10}$ loop between β9-β10 that is directed deep into the Rhs1CT$_{Db10}$ cleft (Fig. 2b). The tip of this inhibition loop forms a tight turn and presents four residues, Asp129-Pro130-His131-Asn132, that interact in a highly specific manner with residues in the Rhs1CT$_{Db10}$ cleft to provide an effective block of the active site. Central to this is RhsI1$_{Db10}$ His131, which participates in π-stacking interactions with Phe1386 of Rhs1CT$_{Db10}$ on one side and van der Waals interactions with Tyr1359 and Val1409 on the other. The imidazole ND1 and NE3 form hydrogen bonds with solvent networks that bridge the partners. The side chain of Asp129 accepts a hydrogen bond from the His131 amide which helps to hold it in place to also accept a hydrogen bond from the hydroxyl group of Rhs1CT$_{Db10}$ Tyr1359, serving to position this aromatic side chain to interact with the imidazole of His131. The RhsI1$_{Db10}$ Pro130 abuts the three aromatic side chains of His131, Tyr1359 and Phe1386. The functional groups on Asn132 participate in solvent mediated links to the partner, in addition to a direct hydrogen bond between ND2 and the carbonyl of Leu1410. The positioning of RhsI1$_{Db10}$ His131 in the cleft of Rhs1CT$_{Db10}$ also provides insight into the catalytic mechanism of the effector (see below). The central role for His131 of RhsI1$_{Db10}$ might suggest it is essential for Rhs1CT$_{Db10}$-RhsI1$_{Db10}$ interaction and toxin neutralisation. On the other hand, the nature of the interface between the two proteins, detailed above, implies that the interaction is unlikely to depend on any single residue and incorporates intrinsic robustness (belt and braces) to guarantee protection against the toxin. In support of the latter scenario, which we believe is likely to be common for effector-immunity interactions, mutation of His 131 to alanine did not prevent

RhsI1$_{Db10}$ from being able to neutralise Rhs1CT$_{Db10}$ activity (Supplementary Figure 2).

## Rhs1CT$_{Db10}$ is an NAD(P)$^+$ glycohydrolase
The most closely related proteins to Rhs1CT$_{Db10}$ identified in the Protein Data Bank (PDB) were the two T6SS-dependent NAD(P)$^+$ glyco-hydrolase toxins, Tne2$_{CT}$ from *P. protegens* Pf-5 (PDB 6B12; Z-score 11.1 and r.m.s.d. of 1.69 Å, over 101 and 95 equivalent Cα positions, respectively) and Tse6$_{CT}$ from *P. aeruginosa* PAO1 (PDB 4ZV0; Z-score 9.2 and r.m.s.d. of 2.48 Å, over 93 and 95 equivalent Cα positions, respectively)[6,7]. In addition, a well-characterised fungal surface NADase, *Af*NADase from *A. fumigatus*[18], was identified that, although limited in overall similarity, provides important details related to enzyme mechanism (PDB 6YGF; Z-score 7.6 and r.m.s.d. of 2.18 Å, over 94 and 92 equivalent Cα positions, respectively). These observations suggested that Rhs1CT$_{Db10}$ might be an NADase toxin.

In order to demonstrate NADase activity in vitro, we required purified Rhs1CT$_{Db10}$. However, it proved impossible to produce Rhs1CT$_{Db10}$ in the absence of RhsI1$_{Db10}$ due to its toxicity towards the producing *E. coli* cells. Therefore, we adopted the approach of co-expressing and co-purifying His$_6$-Rhs1CT$_{Db10}$ with RhsI1$_{Db10}$, followed by denaturation of the His$_6$-Rhs1CT$_{Db10}$-RhsI1$_{Db10}$ complex with 8 M urea or 6 M guanidinium hydrochloride and subsequent on-column refolding of His$_6$-Rhs1CT$_{Db10}$ (described fully in Materials and Methods). In order to determine whether Rhs1CT$_{Db10}$ has NADase activity, we incubated purified His$_6$-Rhs1CT$_{Db10}$ with the potential substrates NAD$^+$, NADP$^+$, NADH and NADPH, then separated the reaction products by HPLC and compared their retention times with standard compounds (Fig. 3). Incubation of His$_6$-Rhs1CT$_{Db10}$ with NAD$^+$ resulted in hydrolysis of NAD$^+$ to ADP-ribose and nicotinamide, and incubation with NADP$^+$ also resulted in production of nicotinamide. No standard compound for pADP-ribose was available but a product peak slightly

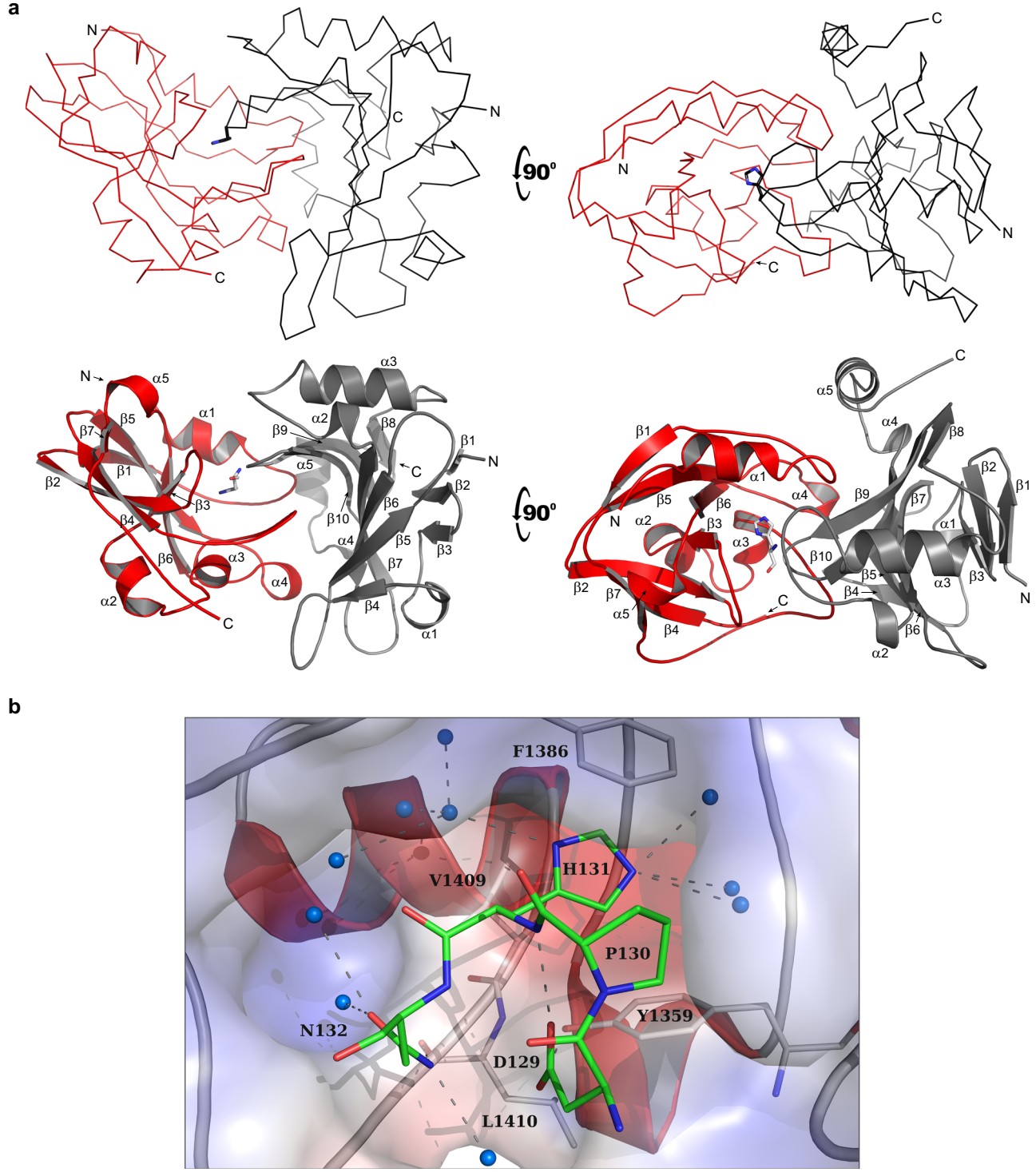

**Fig. 2 | The structure of the Rhs1CT$_{Db10}$-RhsII$_{Db10}$ effector-immunity complex reveals that an inhibition loop of RhsII$_{Db10}$ protrudes into the catalytic cleft of Rhs1CT$_{Db10}$. a** Overall arrangement of the Rhs1CT$_{Db10}$-RhsII$_{Db10}$ complex, showing two views with ribbon (top) and cartoon (bottom) representations of the hetero-dimer. The effector domain (Rhs1CT$_{Db10}$) and immunity protein (RhsII$_{Db10}$) are coloured in red and black, respectively. His131 of RhsII$_{Db10}$ is shown as a stick.

**b** Close-up depiction of immunity protein residues positioned within the putative active site of the effector. Shown is a surface charge representation of the putative active site of Rhs1CT$_{Db10}$ with positively charged groups in red and negatively charged groups in blue. Residues of interest are shown as sticks, with Cα of the effector shown in grey and immunity in green. Water molecules are shown as cyan spheres and hydrogen bonds (within 2.5 Å–3.5 Å distance) as dashed lines.

displaced from the NADP$^+$ substrate peak is presumed to represent pADP-ribose in the latter reaction. In contrast, no activity of His$_6$-Rhs1CT$_{Db10}$ was observed against NADH or NADPH, with HPLC profiles following incubation of His$_6$-Rhs1CT$_{Db10}$ with these substrates being indistinguishable from no-protein controls (Fig. 3a). Confirming the specificity of the reaction, inclusion of RhsII$_{Db10}$, refolded separately

from the His$_6$-Rhs1CT$_{Db10}$-RhsII$_{Db10}$ complex, prevented hydrolysis of NAD(P)$^+$ (Fig. 3b, c). This direct neutralisation of Rhs1CT$_{Db10}$ activity by RhsII$_{Db10}$ is consistent with its occlusion of the predicted active site of the toxin (Fig. 2). These data confirm that Rhs1CT$_{Db10}$ is an NAD(P)$^+$-glycohydrolase able to deplete the essential cellular cofactors NAD$^+$ and NADP$^+$.

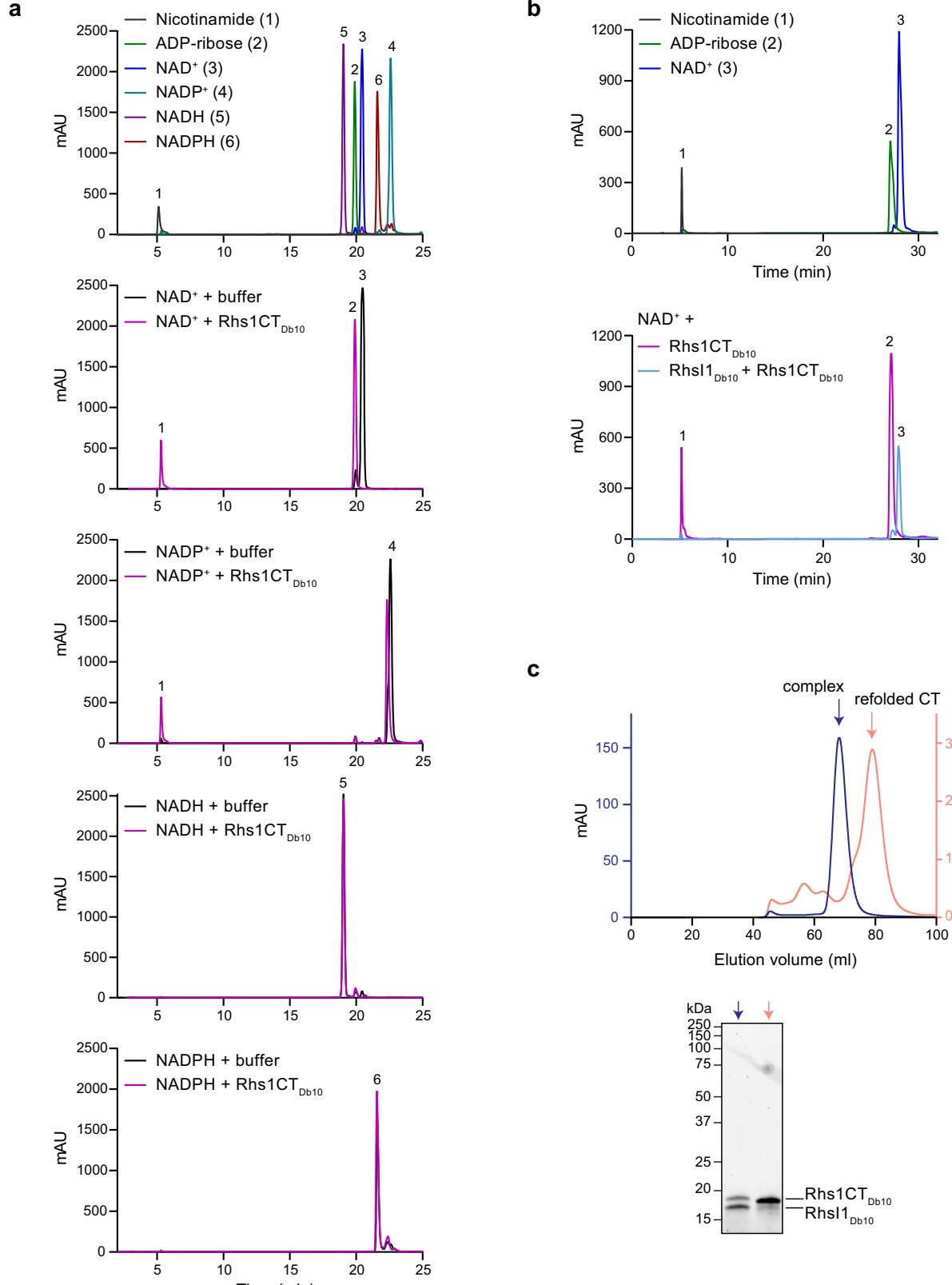

**Fig. 3 | Rhs1CT$_{Db10}$ displays NAD(P)$^+$ glycohydrolase activity in vitro. a** HPLC analysis of standard compounds (nicotinamide, ADP-ribose, NAD$^+$, NADP$^+$, NADH and NADPH; top panel) and of the products following incubation of His$_6$-Rhs1CT$_{Db10}$ with NAD$^+$, NADP$^+$, NADH or NADPH (remaining panels, top-bottom as indicated). Control reactions contained no protein (+buffer). **b** HPLC analysis of standard compounds (nicotinamide, ADP-ribose and NAD$^+$; top panel) and of products following incubation of NAD$^+$ with His$_6$-Rhs1CT$_{Db10}$ alone or with His$_6$-Rhs1CT$_{Db10}$ and RhsI1$_{Db10}$, where RhsI1$_{Db10}$ was added to the substrate prior to His$_6$-Rhs1CT$_{Db10}$ (bottom panel). Colours assigned to each trace are shown in the inset keys. **c** Size exclusion chromatography profiles of the initial His$_6$-Rhs1CT$_{Db10}$-RhsI1$_{Db10}$ complex (dark blue) and refolded His$_6$-Rhs1CT$_{Db10}$ (salmon pink). Proteins were separated using a Superdex 75 16/600 column (top) and peak fractions analysed by SDS-PAGE (bottom). Representative of more than three independent purifications. Source data are provided as a Source Data file.

## The structure of Rhs1CT$_{Db10}$ compared with other NAD(P)$^+$ glycohydrolase toxins reveals clues about substrate recognition and mechanism

Having demonstrated that Rhs1CT$_{Db10}$ is an NADase toxin, we compared its structure with those of the other NAD(P)$^+$ glycohydrolase toxin domains, Tne2$_{CT}$, Tse6$_{CT}$ and *Af*NADase (Fig. 4a). In all four proteins the palm domain is conserved, with β-strands particularly well aligned. Differences are primarily limited to the placement of helical segments on the periphery of the active site cleft. Since only the high-resolution structure of the Rhs1CT$_{Db10}$-RhsI1$_{Db10}$ complex was available, but not the effector domain in complex with appropriate ligands, we used structural and sequence comparisons with these other microbial NADases to help inform on aspects of effector specificity and mechanism.

The hydrolytic cleavage of the glycosidic bond between ADP-ribose and nicotinamide in NAD cofactors requires precise placement of the substrate to interact with a nucleophilic hydroxyl. During the catalytic process a cationic intermediate will be generated that supports C-N bond breakage and C-OH bond formation. Of interest with respect to the recognition of substrate by Rhs1CT$_{Db10}$ is that the structure of *Af*NADase was obtained in complex with an NAD(P)$^+$ mimic, benzamide adenine dinucleotide (PDB 6YGG)[18]. A structural overlay with the *Af*NADase ligand complex suggests that Rhs1CT$_{Db10}$ presents potential phosphate interacting residues in Ser1384, His1412 and Arg1418, whilst Phe1386 is positioned to interact with the ribose moiety of NAD(P)$^+$ (Fig. 4b).

The positioning of aromatic Tyr1359 and Phe1386 side chains might provide the electron rich environment able to support the

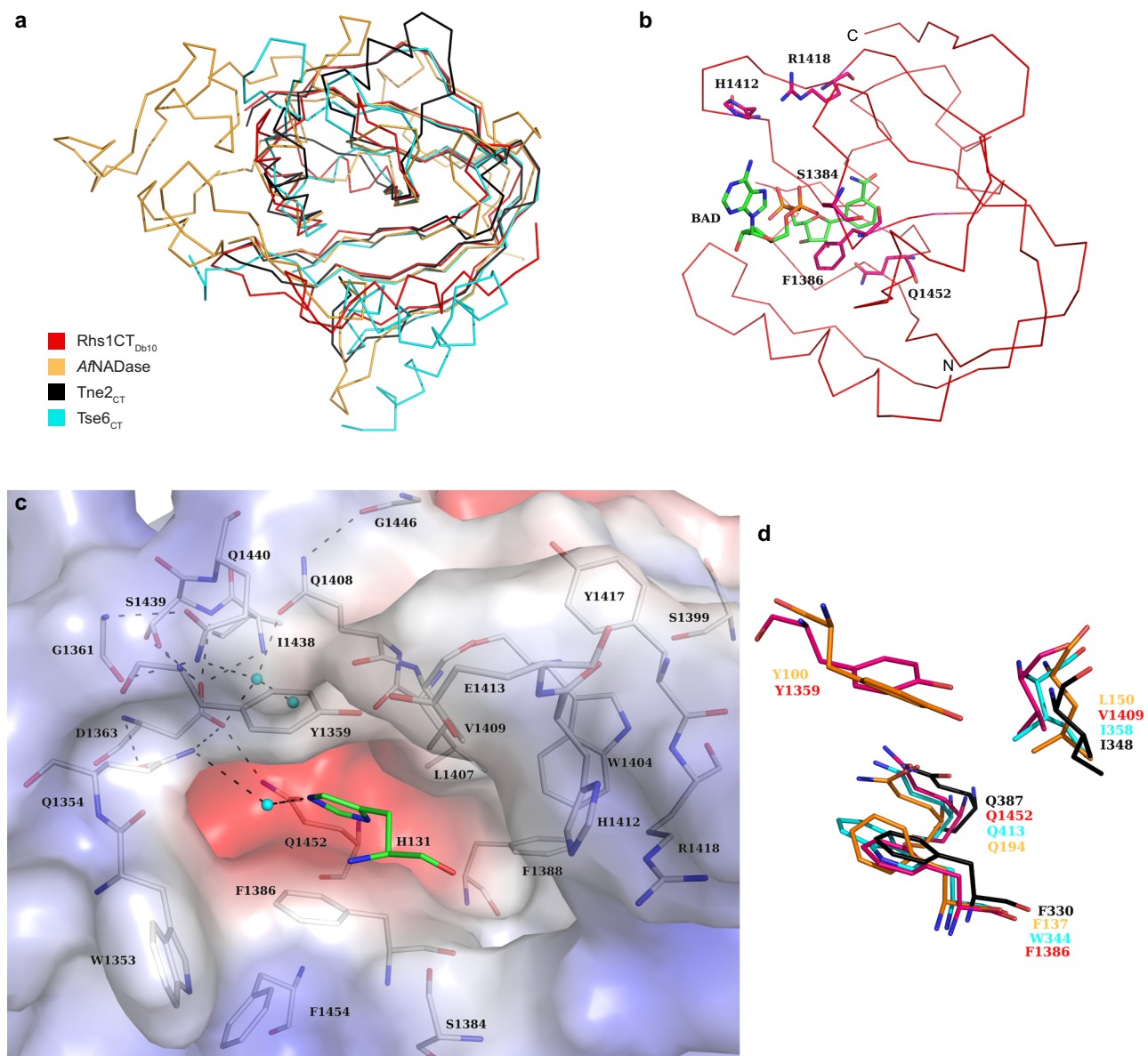

**Fig. 4 | Structural comparison of Rhs1CT$_{Db10}$ with Tne2$_{CT}$, Tse6$_{CT}$ and *Af*NADase informs on enzymatic function and highlights conserved properties of distinct NADase toxins. a** Ribbon overlay of Rhs1CT$_{Db10}$ (red) with known NAD(P)$^+$ glycohydrolase enzymes Tne2$_{CT}$ (PDB 6B12, black), Tse6$_{CT}$ (PDB 4ZV0, cyan) and *Af*NADase (PDB 6YGG, orange). **b** The non-hydrolysable substrate analogue benzamide adenine dinucleotide (BAD) modelled into the active site of Rhs1CT$_{Db10}$ based on its position in the structure of *Af*NADase (PDB 6YGG). **c** Surface charge representation of the putative active site-containing region of Rhs1CT$_{Db10}$ with positively charged groups shown in red and negatively charged groups in blue. Residues of interest are shown as sticks. Residue His131 (in green) is from RhsI1$_{Db10}$. Water molecules are shown as cyan spheres and hydrogen bonds (within 2.5 Å–3.5 Å distance) are shown as dashed lines. **d** Conservation of key substrate-binding residues between the structures compared in panel **a**.

development of a cationic catalytic intermediate (Fig. 4c). It is anticipated that a polar entity would assist the generation of a nucleophile by activating water. Deep on one side of the active site there are three residues, Gln1354, Asp1363 and Gln1452, of interest in this respect (Fig. 4c). The Asp1363 carboxylate, which is furthest away from the catalytic site, is fixed in position by hydrogen bonds accepted from the amide of Ile1438, and the amide and hydroxyl groups of Ser1439. In turn Asp1363 forms hydrogen bonds with Gln1354 and Gln1452 to position those side chains. An ordered water interacts with this glutamine pair, with distances of 2.8 Å and 3.5 Å respectively. This water forms a hydrogen bond, distance 3.1 Å, to His131 NE2 of RhsI1 (Fig. 4c). This polar feature within the cleft may generate the nucleophile or support interactions that correctly position the ribose of the substrate for catalysis to occur in Rhs1CT$_{Db10}$.

Gln1408 and Gln1440 are positioned at the base of the cleft. The side chain of Gln1440 forms two hydrogen bonds with the amide and carbonyl of Gly1361. This positions Gln1440 NE2 to form hydrogen bonds to water that occupies the cleft, near the postulated ribose binding site. The side chain of Gln1408 is directed away from the cleft held down by a hydrogen bond between NE2 and the carbonyl of Gly1446. This positions Gln1408 OE1 to interact with a water that in turn interacts with Asp1363, and another water near the ribose binding site. An alternative rotamer of Gln1408 would position the side chain to stabilise the position of the ribose in the active site. Whilst the mechanism proposed previously[18] is plausible, there remains a question of how the nucleophile is provided. Here, we have identified potential contributors to this aspect of the enzyme mechanism but further work will be required to elucidate it in detail.

The low sequence conservation and structural variation around the catalytic cleft of the four TNT-type NADases which we have compared suggest that whilst a similar reaction is catalysed, aspects of substrate recognition vary. There are however two notable features that are conserved: the placement of aromatic and hydrophobic side chains that appear to bind the reactive part of the substrate exemplified by Tyr1359, Phe1386 and Val1409 in Rhs1CT$_{Db10}$, and a glutamine, Gln1452 in Rhs1CT$_{Db10}$. In the latter case the residue is strictly conserved (Fig. 4d) and appears to be critical for placement of the nicotinamide ribose.

### Investigation of residues potentially important for substrate binding and catalysis of Rhs1CT$_{Db10}$

The structural model indicated which residues of Rhs1CT$_{Db10}$ form the active site and that might contribute to NADase activity (Fig. 4). To investigate the potential contributions of selected residues for Rhs1CT$_{Db10}$ activity, wild type Rhs1CT$_{Db10}$ and mutant derivatives, each with a single candidate amino acid substituted, were produced in *E. coli* under the control of an arabinose-inducible promoter and the ability of each Rhs1CT$_{Db10}$ variant to inhibit growth was assessed. For wild type Rhs1CT$_{Db10}$, no growth was observed on induction with 0.02% or 0.2% L-arabinose. Three variants of Rhs1CT$_{Db10}$ with amino acid substitutions of selected residues predicted to be important for function, namely F1386A, R1418A and Q1452A, showed complete loss of toxicity, indicating that these variants no longer possess NADase activity (Fig. 5a). Three variants of Rhs1CT$_{Db10}$ with substitutions S1384A, S1399A and H1412A, showed a reduction in toxicity, with growth observed on 0.02% L-arabinose but not 0.2% L-arabinose. The structural model suggests that the changes to Ser1399 and His1412, residues distant from the active site cleft, have a minor effect probably due to destabilising the protein by removal of hydrogen bonding interactions. Ser1399 is located at the base of the active site and the side chain accepts a hydrogen bond from the amide of Phe1386, an important residue. Removal of this hydrogen bond might allow Phe1386 a greater conformational freedom leading to a reduction in enzyme efficiency. Such an effect would be expected given that a glycine is placed between the serine and phenylalanine. (Fig. 4c). In order to confirm

that the Rhs1CT$_{Db10}$ variants with reduced or no toxicity were still being produced, Rhs1CT$_{Db10}$ was detected by immunoblotting using a 3xFLAG tag at the N-terminus of the protein. Non-toxic variants F1386A, R1418A and Q1452A were readily detected, confirming production of the protein (Fig. 5b). The variants displaying reduced toxicity, S1384A, S1399A and H1412A, were not detected, similar to wild type Rhs1CT$_{Db10}$, consistent with even the partial activity of these variants being sufficient to disable the cell before high levels of protein can be produced. In the presence of RhsI1$_{Db10}$, all variants were produced at similar levels to wild type Rhs1CT$_{Db10}$ (Fig. 5b). Taken together, these data suggest that, as predicted from our structural analyses, amino acids Phe1386, Arg1418 and Gln1452 are essential for the activity of Rhs1CT$_{Db10}$.

### RhsI1$_{Db10}$ is distinct from other NAD(P)$^+$ glycohydrolase immunity proteins but displays commonalities in inhibition mechanism

The discovery that RhsI1$_{Db10}$ has a previously undescribed fold confirmed that this protein is unrelated to the immunity proteins associated with Tse6$_{CT}$, Tne2$_{CT}$ or TNT and, therefore, represents a distinct family of T6SS immunity determinants from those reported to date. As described above, the structure of the Rhs1CT$_{Db10}$-RhsI1$_{Db10}$ complex identified an inhibition loop in RhsI1$_{Db10}$ which specifically and effectively blocks the active site of Rhs1CT$_{Db10}$ with the key residue His131 (Fig. 2). Further examination of the structure of the complex revealed that the side chain of His131 from RhsI1$_{Db10}$ occupies the site in the toxin where the substrate ribose would be placed. Phe1386 is predicted to form a π-π interaction with the ribose moiety of NAD(P)$^+$ and RhsI1 appears to mimic this interaction as part of the complex formation with the effector by using the imidazole of His131. Although the immunity proteins that bind Rhs1CT$_{Db10}$, Tne2$_{CT}$ and Tse6$_{CT}$, namely Rhs1CT$_{Db10}$, Tni2 and Tsi6, respectively, display distinct folds, they all accomplish enzyme inhibition by occluding the substrate binding site in a stable complex. It is particularly interesting to note that there is a common theme to these interactions. Each immunity protein places a basic residue at the site of catalysis. A structural overlay of the effector proteins, each in complex with their cognate immunity protein, indicates that His131 in RhsI1$_{Db10}$ occupies a similar position to that of Lys62 in Tsi6 and Arg153 in Tni2 (Fig. 6). This similarity supports the idea that mimicking an aspect of the catalytic intermediate or transition state may contribute to the stability of the effector-immunity protein complex.

### Rhs1 in a clinical strain of *S. marcescens*, SJC1036, contains a distinct NAD(P)$^+$ glycohydrolase C-terminal domain

Whilst examining the genome sequences of a large collection of *Serratia* strains[19], we noticed that a clinical isolate of *S. marcescens*, SJC1036, encodes an Rhs1 protein with a CT which might also represent an NADase toxin (Rhs1CT$_{1036}$, amino acids 1333-1486). Sequence-based searches revealed that Rhs1CT$_{1036}$ shares little similarity with Rhs1CT$_{Db10}$ (21% sequence identity), but instead contains the conserved domain associated with TNT (pfam14021, E-value: 3.18 e$^{-34}$). Encoded immediately downstream of Rhs1 in SJC1036 is a small, 124 amino acid protein predicted to be the corresponding immunity protein (RhsI1$_{1036}$). AlphaFold2[20] was used to generate a high confidence structural model of the predicted immunity protein, RhsI1$_{1036}$ (Fig. 7a, right). Inspection of this model in comparison with the available structure of RhsI1$_{Db10}$ revealed that the two structures display different folds and have no apparent structural similarity.

To gain further insight into the function of Rhs1CT$_{1036}$, a high confidence structural model of this effector domain was also generated using AlphaFold2[20] (Fig. 7a, left). Since Rhs1CT$_{1036}$ was predicted to share similarity with TNT and the related NADase *Af*NADase, a structural comparison was performed (Fig. 7b, left). This revealed that the Rhs1CT$_{1036}$ model was highly similar to *Af*NADase (6YGF; 34%

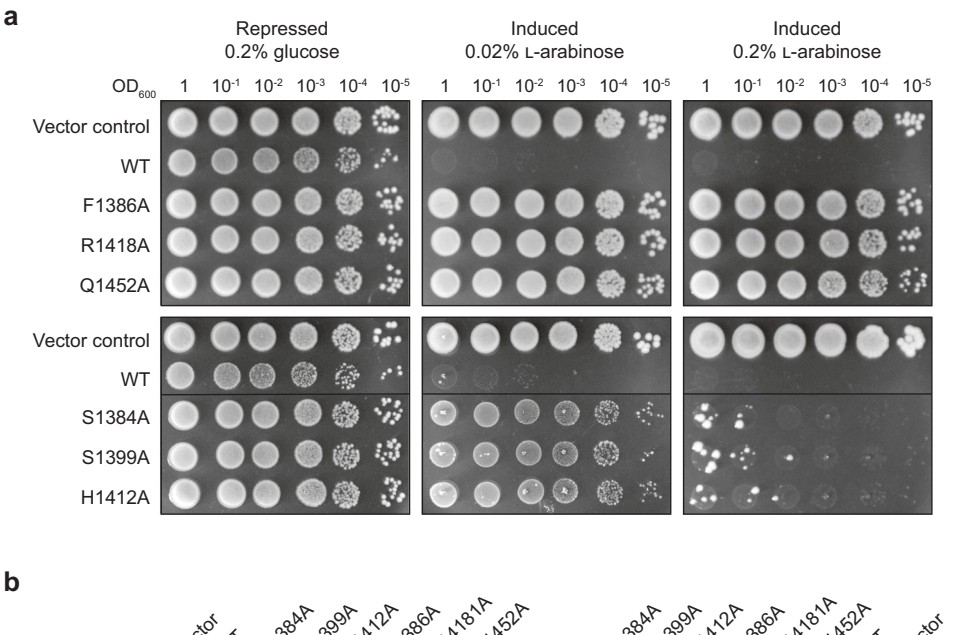

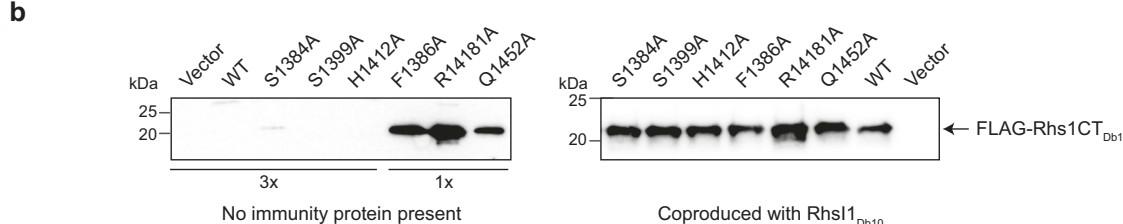

**Fig. 5 | Residues predicted from structural analysis to be important for enzymatic function are required for toxicity of Rhs1CT_{Db10}. a** Growth of *E. coli* MG1655 carrying empty vector control plasmid (pBAD18-Kn) or plasmids directing the expression of wild type Rhs1CT_{Db10} with N-terminal 3xFLAG tag (WT) or derivatives carrying the single amino acid substitutions indicated, on LB media with 0.2% glucose or 0.02% or 0.2% L-arabinose to repress or induce, respectively, gene expression. Representative of two independent experiments. **b** Immunoblot detection of 3xFLAG-tagged Rhs1CT_{Db10} variants following induction with 0.02% L-arabinose in liquid LB media. In the left-hand panel, no immunity protein was present and three-fold more total protein was loaded for the first five samples, as indicated (3x). In the right-hand panel, RhsI1_{Db10} was encoded on the same plasmid and coproduced with each of the 3xFLAG-tagged Rhs1CT_{Db10} variants. Representative of two independent experiments. Source data are provided as a Source Data file.

sequence identity; Z-score 15.9 and r.m.s.d of 0.88 Å, over 107 and 103 equivalent Cα positions, respectively) and also similar to TNT (4QLP; 27% sequence identity; Z-score 10.3 and r.m.s.d of 1.70 Å, over 105 and 102 equivalent Cα positions, respectively). Furthermore, the alignment revealed that residues suggested to be required for binding NAD(P)+ overlapped across the three homologues (Arg1401, Phe1417, Gln1467 of Rhs1CT_{1036}; Arg129, Phe137, Gln194 of *Af*NADase; and Arg757, Tyr765, Gln822 of TNT; Fig. 7c).

Comparing the structures of Rhs1CT_{Db10} and Rhs1CT_{1036} (r.m.s.d of 2.31 Å over 92 equivalent Cα positions), they appear to share a β-sheet core common to all NADases but are otherwise more divergent (Fig. 7b, right). Overall, the structural prediction provided strong evidence that Rhs1CT_{1036} was likely to be an NADase.

In order to determine whether Rhs1CT_{1036} does have NADase activity, we followed the same strategy as for Rhs1CT_{Db10}. Recombinant His_6-tagged Rhs1CT_{1036} was co-produced with RhsI1_{1036}, the complex isolated and His_6-Rhs1CT_{1036} recovered alone by denaturation and refolding. Similar to Rhs1CT_{Db10}, incubation of purified His_6-Rhs1CT_{1036} with NAD+ resulted in its hydrolysis to ADP-ribose and nicotinamide, and incubation with NADP+ also resulted in production of nicotinamide, whilst no activity was observed against NADH or NADPH (Fig. 7d). Therefore, Rhs1CT_{1036} is also an NAD(P)+-glycohydrolase effector.

**The S. marcescens Db10 Rhs1 locus encodes an orphan immunity protein which protects against Rhs1 from SJC1036**

In *S. marcescens* Db10, there are two proteins encoded by genes immediately downstream of *rhsI1_{Db10}* at the 3′ end of the T6SS gene cluster whose function is unknown. However the fact that these genes are co-transcribed with the *rhs1* and *rhsI1_{Db10}* genes[21] implies a functional link with the T6SS. Comparison of the genomic sequences around *rhs1* between Db10 and SJC1036 revealed that the first of these two genes of unknown function, *SMDB11_2280*, encodes a homologue of RhsI1_{1036} from SJC1036 (Fig. 8a, b). Sequence alignment of the corresponding protein, 2280_{Db10}, with RhsI1_{1036} confirmed that the two proteins share 88% identity over their entire length and are almost identical after the first 12 amino acids (Fig. 8c). This suggested that 2280_{Db10} is an orphan immunity protein which might be able to provide protection against Rhs1CT_{1036} or related effectors delivered by competing strains. To test this hypothesis, we first determined whether 2280_{Db10} could provide protection against Rhs1CT_{1036} in an *E. coli* heterologous expression system. As expected, expression of Rhs1CT_{1036} alone resulted in severe inhibition of growth, and this toxicity could be fully alleviated by co-expression of RhsI1_{1036}. Strikingly, co-expression of 2280_{Db10} was also able to provide full protection against Rhs1CT_{1036}, indistinguishable from that of the native immunity protein (Fig. 8d).

Next, we aimed to show that 2280_{Db10} can provide protection to Db10 against Rhs1CT_{1036} in the more relevant context of T6SS-mediated delivery. To do this, we engineered a strain of *S. marcescens* Db10 where *rhs1CT_{Db10}* and *rhsI1_{Db10}* (and the two downstream genes) have been precisely replaced with *rhs1CT_{1036}* and *rhsI1_{1036}*, named Db10 Rhs1_CTI_{1036} (Fig. 8b). In this strain, Rhs1CT_{1036}-RhsI1_{1036} are used in exactly the same way as the native CT-I pair, Rhs1CT_{Db10}-RhsI1_{Db10} (and also the same as they would be in the original background, SJC1036). A control strain, Db10 Rhs1_no CTI, lacking Rhs1CT and all

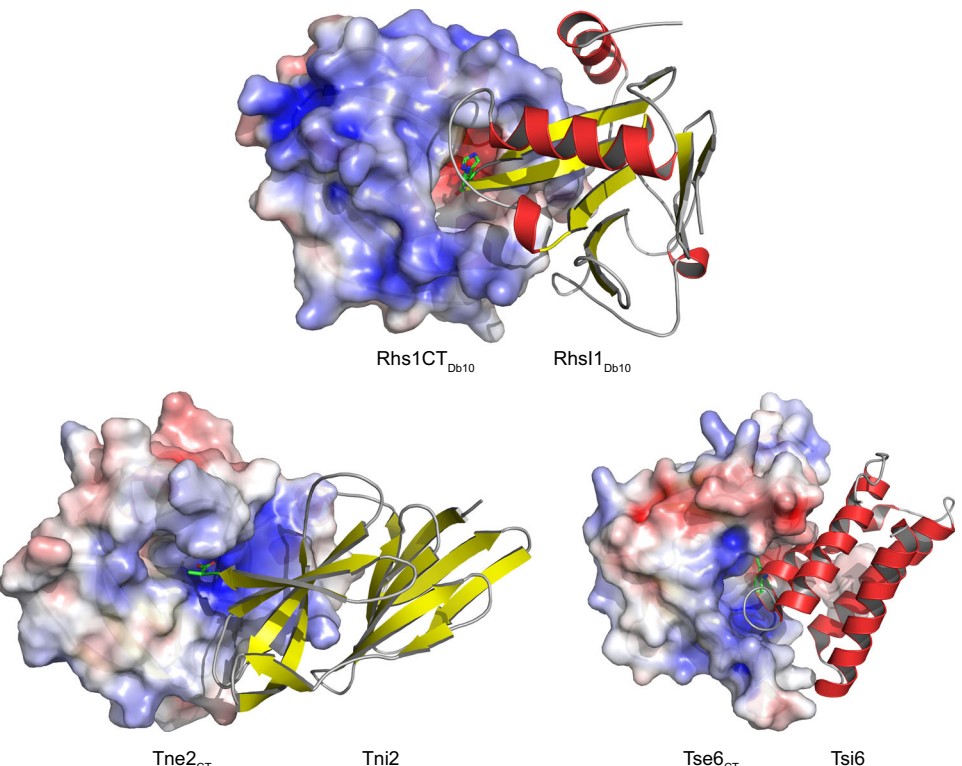

**Fig. 6 | Comparison of effector-immunity interactions between distinct T6SS-associated NADase effectors and their unrelated immunity proteins reveals a conserved inhibition principle.** The structures of three NADase effector-immunity complexes, Rhs1CT_{Db10}-RhsI1_{Db10}, Tne2_{CT}-Tni2 (PDB 6B12) and Tse6_{CT}-Tsi6 (PDB 4ZV0), are shown with their active sites oriented similarly. The effector proteins are shown in surface charge representation, with positively charged groups in red and negatively charged groups in blue. The immunity proteins are shown in cartoon representation, with helices in red, strands in yellow and loops in grey. Basic amino acid side chains of His131, Arg153 and Lys62 from RhsI1_{Db10}, Tni2 and Tsi6, respectively, are inserted into the active site of their respective toxins; these residues are shown as sticks. In the case of Tni2, the side chain of the C-terminal Arg153 was assigned during the current study using the data in PDB 6B12.

three downstream genes was also constructed (Fig. 8b). The Db10 Rhs1_CTI_{1036} strain (attacker) was co-cultured with target strains of Db10 encoding 2280_{Db10} (wild type Db10) or lacking 2280_{Db10} (Db10 Δ2280_{Db10}), and the recovery of the target cells enumerated. If 2280_{Db10} provides protection against toxicity mediated by Rhs1CT_{1036}, recovery of the Δ2280_{Db10} target will be reduced compared with that of the wild type parental strain, which should be fully resistant to Db10 Rhs1_CTI_{1036}. We found that when Db10Δ2280_{Db10} was used as the target strain, there was a reduction in target cell survival when co-cultured with Db10 Rhs1_CTI_{1036} compared with when the attacker lacked Rhs1CT_{1036} (Db10 Rhs1_no CTI) or lacked a functional T6SS (ΔtssE Rhs1_CTI_{1036}). Furthermore, when the engineered Rhs1 was introduced into a genetic background where every T6SS must incorporate and deliver Rhs1 (Δrhs2ΔvgrG1), inhibition of Δ2280_{Db10} was increased further (Fig. 8e). In contrast, there was no difference in the recovery of the parental strain encoding 2280_{Db10} whether co-cultured with attackers able to deliver Rhs1_CTI_{1036} or not. Therefore, 2280_{Db10} is indeed an orphan immunity protein able to provide Db10 with protection from competitors deploying Rhs1CT_{1036}, whilst in the absence of 2280_{Db10}, Db10 is susceptible to this TNT-like NADase effector. These data also confirm that, as expected, RhsI1_{Db10} (which is still present in Δ2280_{Db10}) cannot protect against Rhs1_CT_{1036} (and, similarly, 2280_{Db10} cannot protect against Rhs1_CT_{Db10} since an ΔrhsI1_{Db10} immunity mutant is susceptible to Rhs1_CT_{Db10}[16]).

We noticed that a homologue of RhsI1_{1036}/2280_{Db10} is also encoded downstream of an intact Rhs1-RhsI1 pair at the 3' end of the T6SS gene cluster in *Serratia ficaria* 1D1416, suggesting that this is also an orphan immunity protein protecting against Rhs1CT_{1036}-like NADases (Fig. 8a). Interestingly, the gene encoding the RhsI1 protein in this strain, which presumably protects against the preceding Rhs1CT of unknown function, appears to be homologous with *SMDB11_2281*, the second gene downstream of *rhsI1* in Db10. This strongly suggests that SMDB11_2281 is also an orphan immunity protein, as well as showing that possession of an orphan immunity protecting against Rhs1CT_{1036}-like NADase effectors is not unique to Db10.

## Discussion

Rhs proteins are large polymorphic toxins that are widespread in Gram-negative bacteria and frequently associated with the T6SS. T6SS-associated Rhs proteins contain a highly variable C-terminal effector domain (CT), a shell-like structure which forms around the effector domain, and an N-terminal PAAR-containing domain with a functional role within the T6SS machinery[11,22]. This N-terminal domain includes a PAAR-domain which forms the tip of the expelled puncturing structure and often also transmembrane helices implicated in facilitating movement of the CT across the inner membrane of target cells[12]. Smaller, non-Rhs PAAR-containing effectors may have similar N-terminal PAAR and transmembrane helix-containing domains, which are bound and stabilised by related EagR-family chaperones in both cases, but they do not include the large shell domain of Rhs proteins[12]. Non-Rhs PAAR-containing effectors with NADase CTs have been reported previously, exemplified by Tse6 (renamed Tne1) and Tne2[6,7]. In this study, we have shown that Rhs proteins can also incorporate CTs with NAD(P)+ glycohydrolase activity and have identified two distinct examples of such NADase CTs in one bacterial species.

Rhs1CT_{Db10} and Rhs1CT_{1036} are not closely related to Tne1/Tse6 or Tne2 effectors and appear to represent a distinctive group of Tne (**T**ype VI secretion **N**ADase **e**ffector). Whilst Rhs1CT_{Db10} is more closely

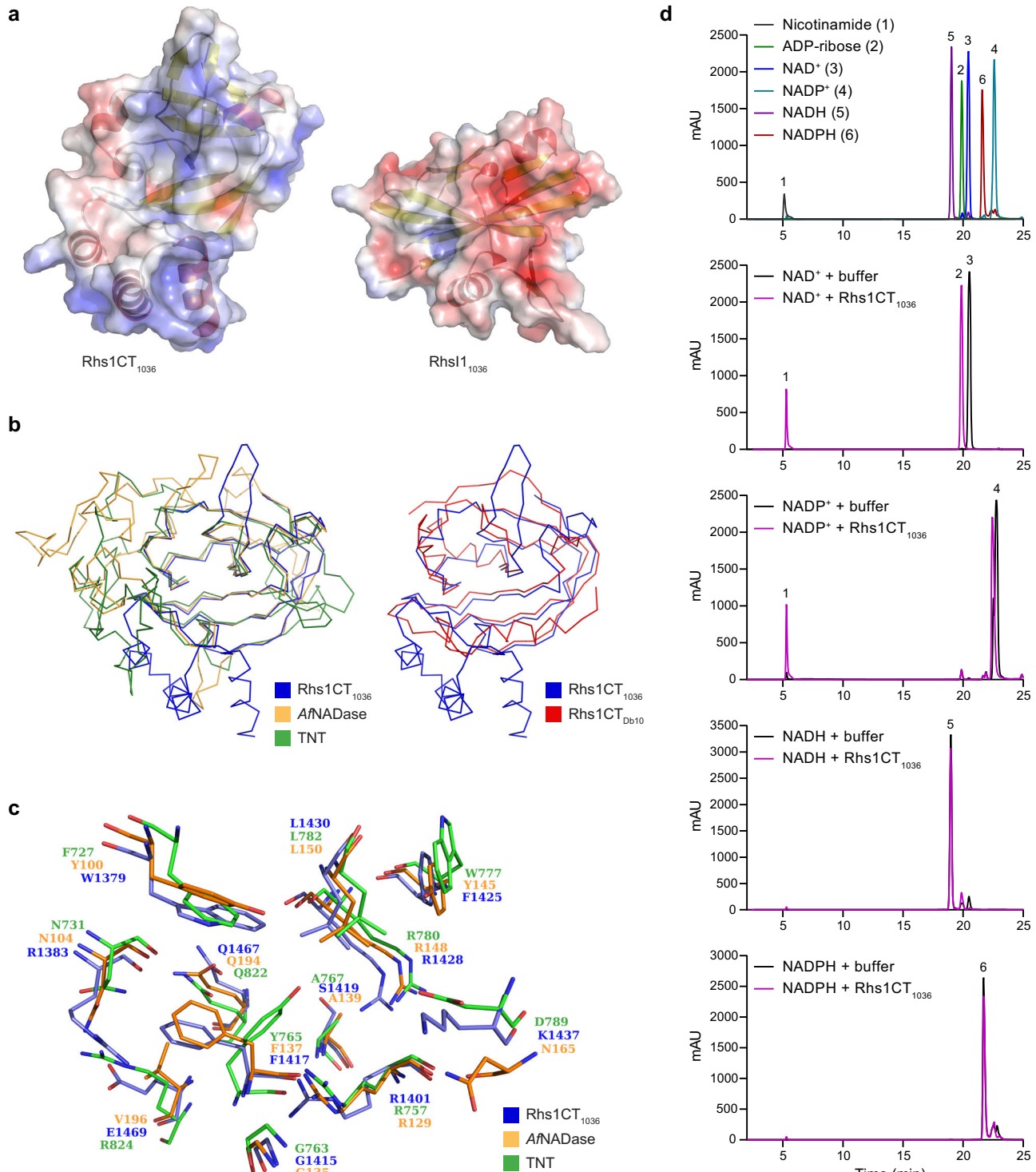

**Fig. 7 | The C-terminal domain of Rhs1 from *S. marcescens* SJC1036 is a TNT-like NAD(P)$^+$ glycohydrolase. a** Models of Rhs1CT$_{1036}$ and RhsI1$_{1036}$ generated using AlphaFold2 and depicted using ribbon overlaid with surface charge representation (positive, red; negative, blue). pLDDT values > 90% and PAE < 1 Å were observed across all residues modelled. **b** Alignments of the predicted structure of Rhs1CT$_{1036}$ (blue) with the structures of TNT (PDB 4QLP, green) and *Af*NADase (PDB 6YGG, orange), left, and Rhs1CT$_{Db10}$ (red), right. **c** Conservation of predicted NAD$^+$-binding and other active site residues (shown in stick representation) between Rhs1CT$_{1036}$, TNT and *Af*NADase. **d** HPLC analysis of standard compounds (nicotinamide, ADP-ribose, NAD$^+$, NADP$^+$, NADH and NADPH; top panel) and of the products following incubation of 500 μg His$_6$-Rhs1CT$_{1036}$ with 5 mM NAD$^+$, NADP$^+$, NADH or NADPH (remaining panels, top-bottom as indicated). Control reactions contained no protein ( + buffer). Colours assigned to each trace are shown in the inset keys.

related to Tse6$_{CT}$ and Tne2$_{CT}$ than other families of NADase toxins at a structural level, the sequence conservation is low (similar to that between Tne1/Tse6$_{CT}$ and Tne2$_{CT}$), and several of the key amino acids predicted to be involved in substrate binding or catalysis are not identical between Rhs1CT$_{Db10}$ and Tse6$_{CT}$ and/or Tne2$_{CT}$. Additionally, the immunity proteins of Rhs1CT$_{Db10}$, Tse6$_{CT}$ and Tne2$_{CT}$ are unrelated. On the other hand, Rhs1CT$_{1036}$ is more closely related to TNT and *Af*NADase NADases, although again there are some differences in

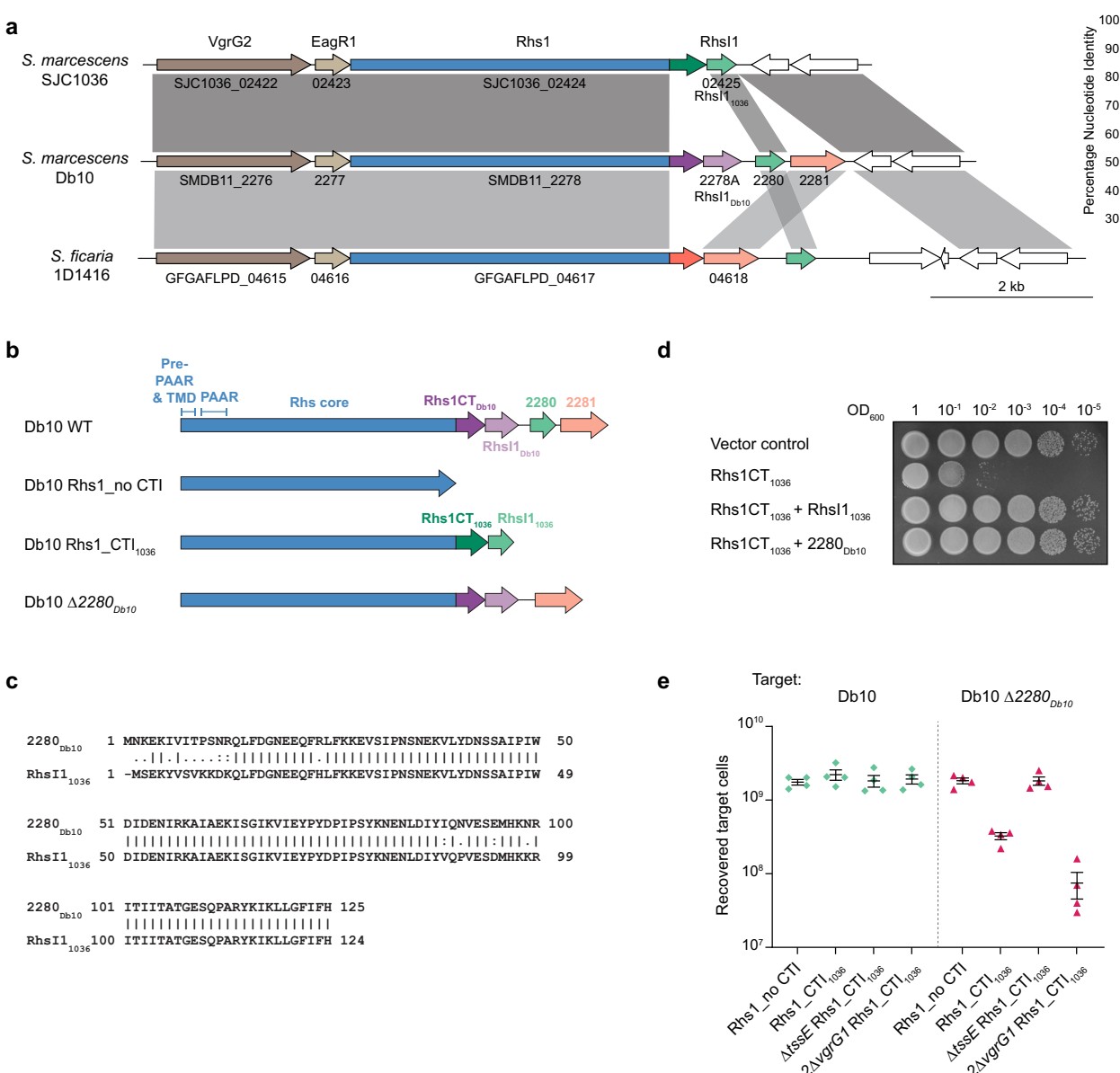

**Fig. 8 | SMDB11_2280 is an orphan immunity protein encoded in the T6SS gene cluster of *S. marcescens* Db10 which is able to provide protection against Rhs1CT_{1036}. a** Synteny plot comparing the 3' end of the T6SS gene clusters of *S. marcescens* SJC1036, *S. marcescens* Db10 and *S. ficaria* 1D1416. Grey shading shows pairwise percentage nucleotide identity as indicated by the key and colours indicate genes encoding homologous proteins. **b** Schematic depiction of the *rhs1* locus in wild type and engineered strains of *S. marcescens* Db10, showing the Rhs protein domains and immunity proteins present in each case and coloured as in (**a**). **c** EMBOSS Needle pairwise amino acid sequence alignment of SMDB11_2280 (2280_{Db10}) and RhsI1_{1036}. **d** Growth of *E. coli* BL21(DE3) pLysS carrying empty vector control (pET15b) or plasmids directing the expression of RhsI1_{1036} alone or

RhsI1_{1036} co-expressed with RhsI1_{1036} or Db10_{2280}, on M9 media with 25 µM IPTG. Representative of two independent experiments. **e** Recovery of parental *S. marcescens* Db10 (green diamonds) and a mutant carrying an in-frame deletion of *SMDB11_2280* (Db10 Δ*2280*, red triangles) as target strains, following co-culture with attacking strains of Db10 for 7.5 h at an initial ratio of 1:1. The attacking strains of Db10 carried engineered *rhs1* loci to direct the expression of Rhs1 with no CT or immunity proteins, or with CT and RhsI from SJC1036 (Rhs1_no CTI and Rhs1_CTI_{1036}, respectively), as depicted in (**b**). Additional deletions generated an inactive T6SS (Δ*tssE*) or a background where all functional T6SSs incorporate Rhs1 (Δ*rhs2*Δ*vgrG1*). Individual data points are overlaid with the mean +/- SEM (*n* = 4 biological replicates). Source data are provided as a Source Data file.

predicted key amino acids and the TNT immunity protein is not related to RhsI1_{1036}. Therefore, Rhs1CT_{Db10} and Rhs1CT_{1036} may represent additional sub-groups, Tne3 and Tne4, of T6SS-delivered NADase effectors. Despite the existence of distinct Tne groupings, it is interesting to note that all have very similar biochemical activity. Rhs1CT_{Db10} and Rhs1CT_{1036}, just like Tne1/Tse6_{CT}, Tne2_{CT}, TNT and *Af*NADase[6,7,18,23], are able to hydrolyse the oxidised cofactors NAD(P)$^+$ but not the reduced forms NAD(P)H. The advantage of this pattern of specificity is not yet clear, but perhaps it represents an effective way to

disrupt cellular NAD homeostasis, specifically the NAD(P)$^+$/NADH ratio, and redox balance.

The existence of four distinct Tne groups, with likely more to be discovered, suggests that disruption of cellular NADase levels is an effective means to intoxicate a rival bacterial cell. This idea is further supported by the presence of Tne2-like toxin domains in putative effectors of the Type VII secretion system (T7SS) used for interbacterial competition in Gram-positive bacteria[7], whilst the antibacterial T7SS of *Streptococcus intermedius* can also secrete a TNT-

like NADase named TelB[24]. Tne2-like effector domains have also been identified in Rhs proteins[7,25], although they have not been studied experimentally. NAD is an essential cellular cofactor involved in numerous and critical redox reactions and a steady level is critical for redox and energy homeostasis in the cell. NAD is one of the most abundant molecules in the cell, but whilst its total levels are high, much of it is bound within proteins and the amount of free NAD available is much lower, meaning that an enzyme able to cleave $NAD(P)^+$ would be able to deplete cellular pools significantly, even if only one molecule is delivered at a time by the T6SS (as is the case for Rhs effectors)[26,27]. Effective depletion of available $NAD(P)^+$ would then likely prevent the cell from being able to maintain its intracellular $NAD(P)^+/NADH$ ratio. Whilst this alone may not be enough to kill the targeted cell or produce the dramatic loss in viable target cell recovery observed with some nuclease effectors[6,14,28,29], it is likely to have a synergistic effect with other effectors simultaneously delivered by the T6SS which damage cellular components or de-energise the cell. Indeed, it has been shown that Tse6 can be strongly synergistic with several other effectors in *P. aeruginosa*, including the membrane depolarising effector Tse4 and peptidoglycan hydrolase Tse1[30]. Further supporting the idea that NADase enzymes may represent a widespread and effective strategy to disable bacterial cells, it has recently been reported that many different phage defence systems use unrelated NADases, containing sirtuin (SIR2) domains, to trigger a process known as abortive infection when infected bacterial cells die or arrest growth without producing phage progeny[31].

Structural analysis of the $Rhs1CT_{Db10}$-$RhsI1_{Db10}$ complex, informed by comparisons with related systems, has advanced our knowledge of effector NADases and of how distinct immunity proteins protect against their potent destructive effect. $Rhs1CT_{Db10}$ possesses a well ordered catalytic cleft, organised by a network of hydrogen bonding interactions and with a distinctive hydrophobic component. Of note, we have identified two key, conserved features of NADase activity that can be ascribed in $Rhs1CT_{Db10}$ to residues Phe1386 and Gln1452. The aromatic residue is placed to support the attraction and binding of substrate, then repulsion of products, whilst the polar residue plays a key role in positioning of substrate for nucleophilic attack. For each NADase effector, the cognate immunity protein positions a basic amino acid at this conserved catalytic centre, mimicking the transition state that would exist during catalysis. Structures of the varied NADase effectors in complex with substrates, products and transition state-based inhibitors, married with detailed kinetic and thermodynamic studies, would, in the future, consolidate our understanding of substrate recognition and mechanism.

This study has also further highlighted the diversity of immunity proteins that are used to neutralise T6SS effector toxins. Whilst all four NADase effector domains have a broadly conserved fold and similar mechanism, the four cognate immunity proteins, $RhsI1_{Db10}$, $RhsI1_{1036}$, Tsi6 and Tni2, display completely distinct folds. Indeed, in this study we show that $RhsI1_{Db10}$ has a previously undescribed protein fold and $RhsI1_{1036}$ is predicted also to adopt a distinct, previously-unreported fold. Consistent with this, $RhsI1_{Db10}$, Tsi6 and Tni2 all have distinct structures and modes of binding to their respective toxins. However, in a lovely example of convergent evolution, all three appear to neutralise the NADase toxin in a similar way. The common mode of inhibition by immunity proteins of NADase effectors relies on the formation of tight, high affinity complex, driven by extensive protein-protein interactions, and the placement of a highly structured loop harbouring a basic amino acid into the active site. This loop serves to occlude the active site from substrate entry whilst the basic residue appears to mimic an aspect of substrate binding and catalytic intermediate formation, namely the presence of a cationic intermediate or transition state. Such a feature may also contribute to the high affinity of the immunity proteins for their targets.

It is common to observe so-called orphan immunity proteins encoded immediately downstream of intact T6SS effector-immunity pairs[1,9,10,32]. These immunity proteins may be homologues of the in-use immunity protein but with a different effector specificity, or they may be from an unrelated family of immunity proteins. It has been widely assumed that such orphan immunity proteins provide protection against incoming effectors which the host strain does not possess, delivered by competitor cells of a different strain or species. However direct experimental evidence for this, particularly in the context of T6SS-mediated delivery, is limited. Here we demonstrate that $2280_{Db10}$ is a *bone fide* orphan immunity protein which is able to protect Db10 against the NADase effector $Rhs1CT_{1036}$ from another strain of *S. marcescens*. $2280_{Db10}$ protects against $Rhs1CT_{1036}$ as effectively as the native immunity protein $RhsI1_{1036}$ in an artificial expression system. Importantly, it also provides Db10 with full protection against T6SS-delivered $Rhs1CT_{1036}$ at native expression levels. Thus, $2280_{Db10}$ is able to protect Db10 from intoxication by other strains of *S. marcescens* which carry a different CT on the same core Rhs protein. This functional demonstration of protection by an orphan immunity protein is consistent with observations by Ross et al. that members of the Bacteroidales possess mobile arrays of genes encoding orphan immunity proteins and that examples of such arrays could provide protection against two effector proteins delivered by the non-canonical Bacteroidales T6SS[10]. It is noteworthy that the presence of genes encoding standalone Rhs immunity proteins to protect against Rhs effectors delivered by other bacteria differs from the current paradigm for Rhs-encoding genetic loci. Previous work on other Rhs effectors, or analogous CdiA polymorphic toxins, has instead reported the accumulation of orphan CT-I (effector-immunity) modules[33,34]. In contrast, we report orphan immunity proteins without a corresponding CT encoded downstream of complete Rhs-RhsI pairs. This highlights an important additional aspect of immunity protein cross-protection and indicates a distinct pattern of CT-I acquisition and exchange in this class of Rhs protein. Interestingly, Rhs immunity proteins may also occur in isolation, as suggested by the recent identification of genes encoding putative Rhs immunity proteins in the absence of any *rhs* effector genes in *Gilliamella*[35].

In this study we have demonstrated that Rhs proteins can be easily re-engineered to deliver and use new CT-I pairs, by replacing the Rhs1CT-I pair of Db10 with that from SJC1036 and demonstrating T6SS-dependent intoxication of parental Db10 by the newly engineered strain. It is believed that RhsCT-I pairs can be readily exchanged in nature via homologous recombination in the highly conserved core Rhs region, often leaving seemingly discarded CT-I pairs immediately downstream of an intact Rhs-RhsI unit[16]. Indeed, such an exchange resulting in a competitive advantage over the parental strain has been demonstrated in an experimental evolution setting[33]. This ability to readily exchange RhsCT-I pairs by homologous recombination helps to explain the huge diversity of Rhs CT-I pairs observed within, as well as between, species, and the corresponding importance of Rhs proteins in both intra-species and inter-species competition. Our engineering approach was designed to mimic such homologous recombination events by precisely replacing the CT-I unit but leaving the remainder of the Rhs1 protein intact (which is almost identical between the two strains). The success of this approach supports its future utility in demonstrating and testing the function of new Rhs CT-I pairs in a physiologically relevant, T6SS-delivery context rather than through their overexpression in isolation. We note the potential of this approach for the development of biocontrol or protein delivery strains designed to deliver specific effector or other protein domains in a biotechnological or therapeutic context.

Finally, we have provided further evidence for the diversity and genetic plasticity of T6SS effector-immunity pairs and orphan immunity genes, supporting the concept of a constant arms race between bacterial strains as they acquire the ability to kill (effectors) and to

resist (immunity proteins) former siblings and new competitors. We have shown that the same toxic enzymatic activity (NADase) can be associated with the T6SS via at least four distinct effector domains. Furthermore, members of the same species can compete against each other using the same activity (NADase), deployed by the same basic effector chassis (Rhs1), through the use of distinct effector domains and immunity proteins. In the case of Db10 and 1036, the arms race has moved a step further as Db10 has acquired or retained an immunity protein ($2280_{Db10}$) able to protect against the 1036-type NADase effector, as well as a further orphan immunity, SMDB11_2281, predicted to protect against an unrelated Rhs1 CT domain. Intriguingly a strain of *S. ficaria* which encodes an intact RhsCT-RhsI pair whose RhsI is homologous with SMDB11_2281, also encodes an orphan RhsI$_{1036}$-like protein which likely protects against Rhs1CT$_{1036}$-like NADase effectors. It seems clear that the arms race is dynamic and complex. It is tempting to speculate that Rhs1CT$_{Db10}$-RhsI1$_{Db10}$ was acquired in a horizontal gene transfer event replacing an Rhs1CT$_{1036}$-RhsI1$_{1036}$-like unit and leaving $2280_{Db10}$ downstream to provide protection against the original strain. However, we do not have the genomic fossil record to ascertain the order of gene acquisition and replacement events in Db10 for certain.

In conclusion, this study has demonstrated the breadth and genetic plasticity of the repertoire of anti-bacterial NADase effectors delivered by the T6SS, as well as providing further insight into their activity and their neutralisation by diverse immunity proteins. We have shown that an orphan immunity protein linked with an Rhs NADase effector can provide full protection against a distinct Rhs NADase effector deployed by another strain during intra-species competition, in addition to demonstrating plug-and-play switching of effector domains between Rhs proteins. These findings support the concept of T6SS-mediated arms races between closely related strains, with the potential to shape a variety of polymicrobial communities.

## Methods

### Bacterial strains, plasmids and culture conditions
Strains and plasmids used in this study are detailed in Supplementary Table 1. Mutant strains of *S. marcescens* Db10 were generated by allelic exchange using the suicide vector pKNG101[21] and streptomycin-resistant derivatives were generated by phage ΦIF3-mediated transduction of the resistance allele from *S. marcescens* Db11[36]. Derivatives of the pRSF Duet-1 and pET15b-TEV plasmids were generated for protein overexpression and purification, whilst plasmids for arabinose- and IPTG-inducible protein expression for toxicity assessment were derived from pBAD18-Kn and pET15b, respectively. Details of oligonucleotide primers or synthetic DNA fragments used in cloning are provided in Supplementary Table 2. Unless otherwise stated, bacterial cultures were grown in LB (LB (10 g L$^{-1}$ tryptone, 5 g L$^{-1}$ yeast extract, 10 g L$^{-1}$ NaCl, with 1.2 g L$^{-1}$ agar for solid media) at 37 °C for *E. coli* and 30 °C for *S. marcescens*. When required, media were supplemented with antibiotics: carbenicillin (Ap) 100 μg ml$^{-1}$, kanamycin (Kn) 50 μg ml$^{-1}$, streptomycin (Sm) 100 μg ml$^{-1}$, chloramphenicol (Cm 25 μg ml$^{-1}$); to maintain repression of proteins expressed from pBAD18-Kn, 0.5% glucose was added to the media for cloning and maintenance.

### Recombinant protein production and purification for crystallography
The genes encoding the effector domain, Rhs1CT$_{Db10}$, and the immunity protein RhsI1$_{Db10}$ were cloned into the two multiple cloning sites of the expression vector pRSF Duet-1 to generate pSC962. This plasmid directs the expression of a stable heterodimeric complex with Rhs1CT$_{Db10}$ carrying an N-terminal hexa-histidine tag (His$_6$). A single colony of freshly transformed *E. coli* BL21(DE3) pLysS was used to inoculate 5 ml of LB media supplemented with Kn and Cm and cultured overnight at 30 °C. This culture was used to inoculate 1 L of the same media and incubated at 30 °C, shaking at 200 rpm, until an OD$_{600}$ of

0.6–0.8 was reached. Expression was induced by addition of IPTG, to a final concentration of 500 μM, and the cultures incubated for a further 3 h. Cells were harvested by centrifugation (1500 g for 20 min) and washed in 50 mM Tris-HCl (pH 8). Cells were flash-frozen in liquid nitrogen and stored at −80 °C until required.

Cell pellets were thawed, resuspended in 10 ml lysis buffer (50 mM Tris-HCl, pH 8.0, 250 mM NaCl, 20 mM imidazole and 5 mM β-mercaptoethanol) supplemented with complete EDTA-free protease inhibitor cocktail (Thermo Scientific) and DNase I (Sigma Aldrich), then passed through an Emulsi-Flex-C3 homogenizer (Avestin). Cell debris were removed by centrifugation at 40,000 g in a Beckman Avanti J-25, (JA 25.50 rotor) for 30 min at 4 °C and the supernatant further clarified through 0.2 μm filters prior to use in affinity chromatography. A 1 mL or 5 mL HisTrap HP (GE Healthcare) column charged with NiCl$_2$ was pre-equilibrated with lysis buffer, and lysate containing His$_6$-Rhs1CT$_{Db10}$-RhsI1$_{Db10}$ was loaded. The column was washed to remove all unbound proteins, and a five-step gradient of lysis buffer supplemented with 500 mM imidazole was applied. The complex eluted at approximately 250 mM imidazole. Fractions containing the complex were identified using SDS-PAGE, pooled and concentrated using a molecular weight cut-off spin concentrator (Millipore).

The protein complex was further purified using size-exclusion chromatography (SEC) on Superdex 75 HiLoad 16/600 column (GE Healthcare). This column was calibrated using molecular weight standards: blue dextran ( > 2,000 kDa), thyroglobulin (669 kDa), ferritin (440 kDa), aldolase (158 kDa), conalbumin (75 kDa), ovalbumin (43 kDa), carbonic anhydrase (29.5 kDa), ribonuclease A (13.7 kDa) and aprotinin (6.5 kDa) (GE Healthcare). The column was pre-equilibrated with two column volumes (CVs) of buffer (50 mM Tris-HCl, pH 8.0, 250 mM NaCl, 1 mM TCEP) and proteins were loaded via 5 ml loop. The sample provided a well-defined profile with an elution volume of about 64.1 ml, corresponding to a molecular mass of ≈ 37 kDa. The theoretical molecular mass of the Rhs1CT$_{Db10}$-RhsI1$_{Db10}$ complex is 35.7 kDa. Fractions containing the complex were pooled and concentrated. SDS-PAGE confirmed the purity of the sample and the presence of two proteins with the mass of 17 and 19 kDa, corresponding to His$_6$-Rhs1CT$_{Db10}$ and RhsI1$_{Db10}$, respectively. The yield of the complex was estimated as ≈ 2.5 mg L$^{-1}$ of *E. coli* culture. Protein concentration was measured in a NanoDrop ND-1000 system (Thermo Scientific) using predicted molar extinction coefficient (ε = 70,360 M$^{-1}$ cm$^{-1}$ at 280 nm) for the complex, obtained from ProtParam[37]. The purified protein complex was then used for crystallography.

Affinity purification and SEC was performed using an ÄKTA pure system equipped with Unicorn 6.4 software (GE Healthcare).

### Crystallisation, X-ray data collection and processing
To identify the lead crystallisation condition, the protein complex was subjected to a range of commercially available screens in a 96-well sitting drop plate format. The first lead crystallisation condition was identified in the JCSG-plus™ screen (Molecular Dimensions) following a two day incubation at 20 °C in drop containing 300 nL of protein complex at 10 mg ml$^{-1}$ and 300 nL of the reservoir solution containing 0.1 M Bis-Tris-HCl pH 5.5, 25% PEG 3350. These crystals were crushed to produce micro seed stocks for optimisation screening in a 24-well sitting drop plate format. Several serial dilutions of seed stocks were prepared, and seeds were introduced to crystallisation drops by passing a nylon loop through a seed stock solution and then dipping it into the crystallisation drops. Crystals with improved shape and dimensions were then observed following two days incubation in a condition containing 2 μl of protein at 10 mg ml$^{-1}$ (50 mM Tris-HCl pH 8.0, 250 mM NaCl, 1 mM TCEP) and 2 μl of reservoir solution (0.1 M Bis-Tris pH 5.5, 25% PEG 3350).

Investigation of the diffraction properties of the crystals together with testing of cryo-protectants and soaking with bromide and iodide was carried out in-house. Crystals were passed through a solution of

mother liquor adjusted to contain 250 mM NaBr, then dipped in PEG400 as a cryoprotectant and immersed and stored in liquid nitrogen. Data were collected at −173 °C, on beamline I03 at the Diamond Light Source (DLS, Didcot, UK) with a wavelength of 0.9150 Å, on the high energy side of the Br K-absorption edge, and an Eiger2 XE 16 M detector. Approximate anomalous scattering contributions at this wavelength are $f'$ −8.5 e⁻ and $f''$ 3.8 e⁻. The data were processed via the automated processing pipeline integrated with XDS[38] and scaled in Aimless[39].

## Structure determination and refinement

The crystal displayed space group $P1$ with unit cell dimensions a = 39.66 Å, b = 44.11 Å, c = 46.81 Å, a = 101.20° b = 96.13° g = 114.15°. A Matthews coefficient of 2.01 Å³ Da⁻¹ suggested a heterodimer in the asymmetric unit with solvent content of around 40% by volume. Initial phases were calculated using the Crank2 experimental phasing pipeline[40], which located seven potential bromide atoms with over 25% probability cut off. Six of these were subsequently included in the refined model. The resulting electron density map at a resolution of 1.3 Å was of excellent quality and the first model was constructed using Buccaneer[41]. The $R_{work}$ and $R_{free}$, the latter based on 5% of the data, were 0.3007 and 0.3276, respectively at this stage.

Rounds of electron and difference density map inspection, model manipulation in COOT[42] and refinement in REFMAC5[43] led to a model consisting of residues Asn1342 to Leu1473 and Met1 to Tyr163 for Rhs1CT_{Db10} and RhsI1_{Db10} respectively. $B$-factors were refined anisotropically and hydrogen atoms were included. Water molecules were assigned to well-defined peaks in the difference density map ( > 3.5 s) that were within 2.5–3.5 Å distance from hydrogen bond donor and acceptor groups. Bromide ions, a molecule of Bis-Tris, and dual rotamers for several amino acid side chains were also included. MolProbity[44] was used in combination with the validation tools provided in COOT to monitor model geometry during refinement. Crystallographic statistics are presented in Supplementary Table 3 and a portion of the electron density map is shown in Supplementary Figure 3. The coordinates and structure factors have been deposited with the Protein Data Bank under accession code 6XTD.

## Structure prediction and in silico analysis

Structural and sequence comparisons were carried out using COOT[42], Dali[45] and XtalPred[46]; structural similarity values are given as Z score (Dali) and r.m.s.d. (generated by COOT v0.9.6). PDBePISA[17] was used to analyse the surface interface between the effector and immunity protein. Molecular images were rendered using PyMOL v2.5.2 (Schrödinger). AlphaFold2[20,47] provided by Colab notebook (Google) was used to generate five prediction models from the query amino acid sequence and a model with the lowest PAE (Predicted aligned Error) and highest pLDDT (per-residue Local Distance Difference Test)[48] scores was selected for subsequent analysis. Genomic synteny plot was generated using genoPlotR v0.8.11 and R v4.0.3.

## Protein unfolding and refolding for isolation of individual components of RhsCT-RhsI complexes

Cleared cell lysates from cultures producing the His_6-Rhs1CT_{Db10}-RhsI1_{Db10} complex from pSC962, or the His_6-Rhs1CT_{1036}-RhsI1_{1036} complex from pSC981, were generated and loaded on a 5 ml His Trap™ HP column as described above. The column was washed with 10 CV of lysis buffer (50 mM Tris-HCl pH 8.0, 250 mM NaCl, 20 mM imidazole, 5 mM β-mercaptoethanol) to remove unbound protein and then with 5 CV of 8 M urea to unfold the complexes and remove unbound immunity proteins. This was followed by a linear gradient from 100% 8 M urea to 100% buffer A over 10 CV. Bound proteins were then eluted with buffer B (50 mM Tris-HCl pH 8.0, 250 mM NaCl, 500 mM imidazole, 5 mM β-mercaptoethanol) and subjected to SEC using Superdex 75 HiLoad 16/600 column (equilibrated in 50 mM Tris-HCl, pH 8.0, 250 mM NaCl, 1 mM TCEP) to separate out the refolded toxin from any

remaining effector-immunity complex that was not refolded. In the case of the experiment shown in Fig. 5b, unfolding was performed by incubating the protein complex in 5 M guanidinium hydrochloride at 90 °C for 5 min and then performing on-column refolding as described above. The immunity protein was recovered by dialysing the flow-through containing guanidinium hydrochloride against fresh buffer containing 25 mM Bis-Tris-HCl pH 6.0, 25 mM imidazole, 250 mM NaCl and 2 mM DTT.

## NADase assays

Unless stated otherwise, reactions were prepared in 50 μl buffer (50 mM Tris-HCl pH 8.0, 250 mM NaCl) supplemented with 5 mM substrate (NAD⁺, NADH, NADP⁺, or NADPH), followed by addition of 500 μg purified Rhs1CT_{Db10} or Rhs1_{1036} and incubation at 30 °C for 30 min. Then 50 μl acetonitrile was added to precipitate the protein and the precipitate was removed by filtration through a mini spin column (Neo Biotech). In the case of the experiment shown in Fig. 5b, reactions were performed in 100 μl buffer (50 mM Tris-HCl pH 7.5, 250 mM NaCl, 0.5 mM TCEP) containing 2.5 mM NAD⁺ (Rhs1CT_{Db10} + RhsI1_{Db10}) or 5 mM NAD⁺ (Rhs1CT_{Db10} only), and either 30 μg RhsI1_{Db10} and 15 μg Rhs1CT_{Db10} (with RhsI1_{Db10} premixed with the substrate before the addition of Rhs1CT_{Db10}), or 15 μg Rhs1CT_{Db10} only, were added, followed by incubation overnight at 30 °C. Analysis of standard compounds (5 mM) and reaction mixtures by HPLC was performed using the UltiMate 3000 HPLC system (Dionex) with Chromeleon (v6.8) software and an XBridge BEH-amide column (Waters), using a flow rate of 4 ml min⁻¹. The column was equilibrated and washed between runs in 90% buffer A (95% v/v acetonitrile, 10 mM NH_4CH_3CO_2, pH 8.0) and 10% buffer B (50% v/v acetonitrile, 10 mM NH_4CH_3CO_2, pH 8.0). Compounds were separated using a linear gradient from 10% buffer B to 100% buffer B over 25 min (Figs. 3a, 7d) or 40 min (Fig. 3b) and absorbance was monitored at 280 nm.

## In vivo toxicity assays

Genes encoding wild type 3xFLAG-Rhs1CT_{Db10} or variants with single amino acid substitutions were cloned under the control of an arabinose-inducible promoter in pBAD18-Kan and expressed in E. coli MG1655. Genes encoding Rhs1CT_{1036} with or without RhsI1_{1036} and 2280_{Db10} were cloned under the control of an IPTG-inducible promoter in pET15b and expressed in E. coli BL21 (DE3) pLysS. Cells of freshly transformed E. coli grown overnight on solid media were resuspended in LB or M9 liquid media, normalised to OD_{600} of 1, serially diluted from 10⁰ to 10⁻⁵, and 5 μl of each dilution spotted on LB or M9 agar plates with appropriate supplements. The plates were incubated overnight at 37 °C.

Protein levels of the 3xFLAG-Rhs1CT_{Db10} variants were visualised using an anti-FLAG immunoblot. Cultures of MG1655 carrying the above plasmids were grown overnight in LB + 0.2% glucose, subcultured 75 μl into 5 ml LB + 0.2% glucose and grown for 1.5 h, then subcultured 75 μl into 5 ml LB + 0.02% L-arabinose and grown for 2 h. Finally cells equivalent to 1 ml culture at OD_{600} 1 were recovered by centrifugation, resuspended in 75 μl Laemmli SDS-PAGE sample buffer with β-mercaptoethanol, heated to 100 °C for 10 min. Finally, 5 μl of each sample was subjected to SDS-PAGE and immunoblotting using anti-FLAG primary antibody (Sigma, catalogue number F3165, 1:10,000) and HRP-conjugated anti-mouse secondary antibody (Bio-Rad, catalogue number 170-6516, 1:10,000). Uncropped and unprocessed blot images are supplied in the Source Data File.

## Co-culture assays for T6SS-mediated anti-bacterial activity

Cells of relevant strains of S. marcescens Db10 grown overnight on solid LB media were resuspended in LB and normalised to OD_{600} 0.5. The attacker and target were mixed at a 1:1 ratio and 25 μl of the mixture grown on solid LB at 30 °C for 7.5 h. Following the co-culture, cells were recovered in 1 ml LB and the number of surviving target cells was

enumerated by serial dilution and viable counts on Sm-supplemented LB agar.

## Reporting summary

Further information on research design is available in the Nature Portfolio Reporting Summary linked to this article.

## Data availability

The coordinates and structure factors generated in this study have been deposited in the Protein Data Bank under accession code 6XTD. All other data generated in this study are provided within the paper and its Supplementary Information files. Source Data are provided with this paper. Other structural data used in this study are available in the Protein Data Bank under accession codes 6B12 (Tne2_{CT}-Tni2), 4ZV0 (Tse6_{CT}-Tsi6), 6YGF (*Af*NADase), 6YGG (*Af*NADase in complex with benzamide adenine dinucleotide), and 4QLP (TNT). Bacterial genome sequences used in this study have Genbank accession codes GCA_000513215.1 (*Serratia marcescens* Db11), GCA_946406795.1 (*Serratia marcescens* SJC1036) and GCA_003641105.1 (*Serratia ficaria* 1D1416). Source data are provided with this paper.

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

## Acknowledgements

This work was supported by Wellcome (grant numbers 215599/Z/19/Z, S.J.C & W.N.H; 104556/Z/14/Z, Senior Research Fellowship S.J.C.; 220321/Z/20/Z, Senior Research Fellowship Renewal S.J.C.) and the Medical Research Council (grant number MR/N013735/1, PhD studentship M.H.). The Diamond Synchrotron Light source is acknowledged for beamtime. We would like to thank Kieron Lucas for assistance with HPLC and Paul Fyfe for crystallography support. We would also like to acknowledge Juliana Alcoforado Diniz for the construction of the Δ2280$_{Db10}$ mutant, and Sarah Brough for assisting in constructing pSC3019 and pSC3023 and noting the homology between RhsI1$_{1036}$ and SMDB11_2280. For the purpose of Open Access, the authors have applied a CC BY public copyright license to any Author Accepted Manuscript version arising from this submission.

## Author contributions

M.H., W.N.H., and S.J.C. conceived the study; M.H., G.P., W.N.H., and S.J.C. designed experiments and analysed data; M.H., G.P., R.G.M., C.E., and G.B. performed experimental work; D.J.W. and G.P. performed bioinformatics analyses; G.P., W.N.H., and S.J.C wrote the manuscript with input from the other authors.

## Competing interests

The authors declare no competing interests.
