## [Peer Review File · Nature Communications]

REVIEWER COMMENTS

Reviewer #1 (Remarks to the Author):

Comments to the authors:

In this paper the authors investigate the molecular activity of one of the two Rhs toxins in *Serratia marcescens* DB10. Using structural analysis the authors find that the toxin adopts a fold similar to other T6S effectors with NADase activity. Further they show that the toxin indeed possesses NAD(P)⁺ degrading activity and that the immunity protein neutralizes this activity by inserting a His residue into the active site. This mechanism of neutralization seems to be conserved among T6S effectors with NADase activity. They next identify key residues in the active site and show that mutation of these residues renders the enzyme inactive. Finally, the authors look into the two genes found down-stream of the Rhs toxin and its immunity and find that they represent putative immunity proteins for other Rhs toxins in *Serratia marcescens* 1036 and *Serratia ficaria* respectively. Investigating the 1036 toxin more in detail, they reveal that this also possesses the ability to degrade NAD(P)⁺. Using nifty allelic replacement, they create an inhibitor able to deliver the 1036 Rhs toxin and show that the down-stream gene in DB10 indeed functions as an immunity protein towards the 1036 Rhs toxin. This is the first evidence that bacteria would accumulate Rhs immunity genes to protect against other closely related strains. The work here is very well performed and an important contribution to the field. For the most part the experiments have the appropriate controls, but there are some concerns with one experiment in particular that should be addressed to make the paper to be acceptable for publication. My main issues with the paper is the following:

1. To identify the active site in the Rhs toxin the authors create 6 amino acid substitutions in the Rhs toxin, which abolish or reduce the toxicity. The authors then perform western blots to confirm that these mutant proteins are still being produced, but fail to detect a band for the WT and three of the mutants with reduced activity. For the mutants with no activity, strong bands can be seen in Fig 5b. This is problematic as mutations in a protein could make it inactive not only due to changes in the active site, but because the mutation changes the folding of the protein. To confirm that the mutations introduced here do not change the folding of the protein, the authors should attempt to perform co-immunoprecipitation with the immunity. The immunity will only bind if the toxin is correctly folded. Although the immunity interaction with the active site could be changed by these mutations, it is not likely that a single amino acid substitution would abolish the interaction between the toxin and immunity altogether. Thus, co-IP in the presence of a cross-linker should be able to confirm that the mutations are not affecting the folding of the toxin. At a minimum, a western blot in the presence of immunity should be performed to show that these toxins can indeed be expressed when sufficient immunity is around.

2. On a similar line, the authors suggest that a basic residue in the immunity is important for neutralization of the Tse6, Tne1 and Rhs NADase toxins respectively. An amino acid substitution to replace the His131 residue in the RhsI protein and showing lack of protection would have strongly strengthened their case.

3. The finding that some bacteria accumulate Rhs immunity proteins to protect against other toxins is novel and not previously acknowledged. Previous work on other Rhs toxins or their CdiA homologues instead show an accumulation of orphan toxin-immunity modules, suggesting that something is vastly different for the Rhs toxin-immunity pairs identified here. Mentioning this discrepancy could strengthen the importance of the findings.

Reviewer #2 (Remarks to the Author):

In this study by Hagan et al., the authors investigate Rhs EI pairs involved in inter-bacterial competition in *S. marcescens* strains. It is a well-executed study, with a thorough structural investigation of the EI pair and interesting identification of genes encoding orphan immunity proteins. Although most of the findings in the manuscript have been previously described, the study contributes considerably to the development of knowledge in the field.

There are some general comments and a few minor suggestions for improvement.

1. The authors identified Rhs effectors in two strains that possess NADase toxin domains belonging to different subgroups of this family of proteins. NADase toxins have previously been identified, but it is notable to see that there is a great variety (4 different subfamilies and probably more could be expected) demonstrating their importance for bacterial competition. A phylogenetic analysis of T6SS and non-T6SS NADase toxins would be a beneficial addition to further support the diversity of these toxins and their potential evolutionary relationship.

2. The authors used comparative structural analysis to identify conserved residues required for NADase activity and demonstrate it by inhibition assays. They reveal that, as expected, the different families of NADase used non-related immunity proteins, but interestingly they use a shared mechanism of inhibition. The structural section of the manuscript is very comprehensive and well presented in general, but on some occasions, the information is repetitive and could be condensed and combined to improve readability. That is, Figures 1 and 2 could be combined to avoid overlap. The same applies to Figures 4 and 5. The information from Figure 6 is also redundant in Figure 2. Paragraph 234-252 could be

restructured, as it repeats information from earlier in the manuscript. The content in Lanes 259-264 is repetitive with the text in the following section. Similarly, the content in lanes 386-388 is repetitive with the content in lanes 401-404.

3. The study also shows the exchange of CTs between strains, which has been previously demonstrated, and the authors should cite these works. It is especially relevant that the authors acknowledge the pioneering work of Koskiniemi et al., 2014 where they used a relevant setup to show the selection of an evolved population that undergoes recombination of the CT and the immunity pair (<https://www.ncbi.nlm.nih.gov/pmc/articles/PMC3967940/>).

4. The authors identified orphan T6SS immunity genes downstream of the study EI pair. T6SS orphan immunity genes have been previously identified in other model organisms, but the magnitude of these orphan immunity proteins is not fully understood and is particularly intriguing. This section of the manuscript could be further expanded with an *in silico* study of the overall distribution of these genes. It would be of great interest to see if the identification of genes encoding orphan immunity proteins in *Serratia* is simply a curiosity, or if it is a frequent phenomenon. It is worth exploring whether these genes produce immunity proteins that protect exclusively against Rhs effectors or if it is a general event that includes protection against other types of effectors as well. Likewise, it is of interest to understand if they appear exclusively in *Serratia* or if they can be found in other genres; and if they are found exclusively in T6SS+ bacteria or any bacteria or T6SS-susceptible microorganisms. Among *Serratia* species, is it common or rare to find T6SS orphan immunity genes? All these questions could be further investigated to provide a more comprehensive understanding of the significance and prevalence of orphan immunity genes in T6SS-mediated inter-bacterial competition.

5. The immunity proteins described in the manuscript belong to different families and seem specific to their cognate effectors. It would be of interest to test whether they can protect against NADases from other subfamilies. The phylogenetic tree would provide insight into the closest effectors, and specificity could be tested between effector-immunity pairs belonging to the same or different subfamilies.

6. Rhs effectors have been identified in other delivery systems different from T6SS; the authors should consider discussing these systems further. A relevant example would be described here (<https://journals.plos.org/plosgenetics/article?id=10.1371/journal.pgen.1002217>)

7. The authors commented on the fact that the four identified T6SS NADase hydrolysed the oxidised cofactors but not the reduced forms. They could expand this comment including the behaviour of non-T6SS NADases.

Lane 34: The "the" word is not needed before "it delivers", which would read better as "where it delivers".

Lane 335 and 338. The word 'smaller' could be replaced by 'non-RhS' to refer to PAAR-containing effectors that do not have the core RhS domain.

Lane 464-368. This sentence requires a reference.

Lane 371. Other examples of T6SS nucleases: <https://www.pnas.org/doi/10.1073/pnas.1814181115>;
<https://www.nature.com/articles/ismej2016169>

Lane 405-6. The authors stated that orphan immunity proteins are encoded immediately downstream of intact T6SS EI pairs, but this is not necessarily the case, and arrays of genes encoding immunity proteins can be found even in bacteria without T6SSs. The authors should consider revising this statement and providing more context or examples. Citation number 1 does not seem adequate in this context.

Lane 423-426. This sentence requires a reference.

Lane 427. The importance of RhS effectors is not limited to intra-species competition. The authors should consider generalising this argument.

Lane 440-444. To improve the readability of this statement, the authors should discuss whether these two strains coexist in their natural niche.

Figure 2A. The authors should consider whether the upper panel of this figure is necessary or could go to supplementary material.

Figure 3. The authors should comment on the fact that the same peaks are at 20 min in panel A and at 30 min in panel B.

Reviewer #3 (Remarks to the Author):

This is a very good research work, however the presentation is not very good. It require improvement.

Nature_Comms_413902

This manuscript present results obtained from combinations of 'Theoretical Biology', 'Molecular Biology', 'Structural Biology' and the 'function' with a main emphasis on 'Heterodimer' 3D-structural result of Rhs1CTDb10 - Rhs1Db10. However, it remains unclear until page 8, line 212 that Db10 is the wild type. Moreover, in this manuscript, different results are obtained from different experimental observations and combinations of more than one technique. But at the end, I am not very impressed with the layout, and mode of presentation as:

- i) The results are not well documented.
- ii) It is very difficult to follow the flow of this exciting and wonderful research work.
- iii) Finally, detailing of relevant information is not well covered either.

Lists of my concerns are given bellow –

Comments (C), questions (Q) and criticisms (CS) -

Several abbreviations have been used in this manuscript, so it is better to add the full form and details on their first use (C).

Abstract -

Line (L) 10 – model Db10 or wild type Db10?

3D-structure the heterodimer of wild type strain Db10 – but not clear here.

Was the subsequent comparative analysis of Db10 and SJC1036 done using the model of SJC1036?

Introduction -

Page 3

Line (L) 24 – In my opinion before the beginning of T6SS functions, it would be better to add a few words about the effectors Hcp, PAAR etc. – which would make it easier for the readers to follow the text (C) mentioned on L34.

Page 4

L 66 – 67 – 'The bacterium *S. marcescens* is present in diverse environmental conditions and is responsible in hospital-acquired infections' –

- i) What are these and how severe are these infections?

L 69 – ‘at least ten effector proteins, including two Rhs proteins’ – one is Rhs1, what is the other?

L-81 – ‘orphan immunity protein gene – add the details of this gene here (C).

Page 5

(C) – At the beginning the details of residues, the range of Rhs1CTDb10 - Rhs1Db10 complex should be added. The main reason is that, on this page, lots have been discussed about the structure, however there is no mention about the Mw of this complex and what was the span of residues that it constitute.

L 101 - 102 – Is there any specific nature of the residues, e. g. electro +ve/-ve, acidic etc. which are on top of the putative active site? (Q)

Page 6

L124 -126 – D129 – N132 loop of Rhs1Db10 resulted to active site blockage of Rhs1CTDb10 – hence it can be classified as closed or inactive structure – are there any comments on the opening and closing of the active site?

L 137 – 139 – What is the sequence homology?

L 141 –PDB code of NADase is missing. (In the PDB there are three structures NADase is 6YGE the other two. 6YGF and 6YGG are complex with various ligands.

Page 7

L 175 and 403 – ‘Presence of a Cationic interaction/intermediate or a transition state - - which support breakage of C – N bond and the formation of C - OH bond’ – any experimental evidence in support of this statement? (Q)

Page 8

L 199 - 201 – Either you must propose what are the possible potential contributors to validate the enzyme mechanism or have to remove the entire sentence – (C).

Page 9

L236 – 238 - Rhs1Db10 fold – Is it really a new fold? Could you please check again?

- To me it looks to be a very common 13-sandwich, 13-trefoil and other common 13-sheet dominated protein fold.

- This is only my comments, because I cannot check myself without the coordinate.

L 240 – ‘As observed H131 is the key residue to block the active site’ – was there any attempt to mutate H131?

- If so, what was the observation?

- My own and readers main interest would be to know whether H131 is the key in opening and closing of the active site. In my knowledge, I know few structures where a single residue was key in opening and closing the active site of the structure.

- In others, the key was the residue along with other interactions.

Page 10

Fig 7A – structural model of RhsI11036 – based on which structure the model built? (Q)

- What are the major differences?

- What is the sequence homology of RhsI1Db10 and RhsI11036? (Q)

Fig 7B - Superposition of Rhs1CTDb10 and RhsICT11036 – it looks like Rhs1CT1036 has a deletion peptide – if so, how long is that peptide?

L 290 and Fig 8B – ‘Comparison of Db10 and SJC1036 genome around rhs1 - revealed two genes ---: In Db10 9i) Rhs1db10 (ii) 2280 whereas in SJC1036

RhsI11036 is missing but similar to 2280 another gene is present’ – is that a homologue of RhsI1db10?

It is not at all clear from the text on line 290 - 291.

Page 33 – NADase PDB is not 6YGG rather 6YGE is complex of NADase with various ligands.

Page 11

L 301 -327 – (C)

- Well planned work, however the presentation of this section is not up to the standard; in my opinion this could have been presented much better.

- To me, this section is overwhelmed, congested, and very difficult to follow.

- I would add a small table and less text, to make this section very attractive and easy for the researcher/reader to follow.

Page 12

L 357 – ‘The existence of four distinct Tne groups’ – Tne1 and Tne2 sub-groups have already been well established in T6SS, but not yet for Tne3 and Tne4.

- At this stage, I am reluctant to accept the existence of four distinct sub-groups of Tne, instead there may be two other new sub-groups 3 & 4.

Page 17 – JCSG screen reference is missing.

Page 18 –

L 541 – Asn24 and Leu155 – is not quite right – I think those are Asn1342 and Leu1473?

Page 19 – No Fig 5C

L586 – What is MeCN?

Page 21 – Reference 22 and 23– incomplete.

My decision is – Acceptance and publication of this manuscript subject to clarification and appropriate modifications listed above.

Detailed Response to Reviewers' Comments (Hagan, Pankov *et al.*, NCOMMS-23-06527)

We would like to thank the Reviewers for taking the time to positively and constructively review our manuscript. The manuscript has been revised in response to the comments and suggestions of the Reviewers and has been improved as a result. As requested, the Reviewers' comments have been reproduced verbatim (in black text) and our responses are included below each point (in blue text).

REVIEWER COMMENTS

Reviewer #1 (Remarks to the Author):

In this paper the authors investigate the molecular activity of one of the two Rhs toxins in *Serratia marcescens* DB10. Using structural analysis the authors find that the toxin adopts a fold similar to other T6S effectors with NADase activity. Further they show that the toxin indeed possesses NAD(P)⁺ degrading activity and that the immunity protein neutralizes this activity by inserting a His residue into the active site. This mechanism of neutralization seems to be conserved among T6S effectors with NADase activity. They next identify key residues in the active site and show that mutation of these residues renders the enzyme inactive. Finally, the authors look into the two genes found down-stream of the Rhs toxin and its immunity and find that they represent putative immunity proteins for other Rhs toxins in *Serratia marcescens* 1036 and *Serratia ficaria* respectively. Investigating the 1036 toxin more in detail, they reveal that this also possesses the ability to degrade NAD(P)⁺. Using nifty allelic replacement, they create an inhibitor able to deliver the 1036 Rhs toxin and show that the down-stream gene in DB10 indeed functions as an immunity protein towards the 1036 Rhs toxin. This is the first evidence that bacteria would accumulate Rhs immunity genes to protect against other closely related strains. **The work here is very well performed and an important contribution to the field.**

We thank the Reviewer for their positive comments and are particularly pleased to hear that the Reviewer finds the work to be 'very well performed' and 'an important contribution to the field'.

For the most part the experiments have the appropriate controls, but there are some concerns with one experiment in particular that should be addressed to make the paper to be acceptable for publication. My main issues with the paper is the following:

1. To identify the active site in the Rhs toxin the authors create 6 amino acid substitutions in the Rhs toxin, which abolish or reduce the toxicity. The authors then perform western blots to confirm that these mutant proteins are still being produced, but fail to detect a band for the WT and three of the mutants with reduced activity. For the mutants with no activity, strong bands can be seen in Fig 5b. This is problematic as mutations in a protein could make it inactive not only due to changes in the active site, but because the mutation changes the folding of the protein. To confirm that the mutations introduced here do not change the folding of the protein, the authors should attempt to perform co-immunoprecipitation with the immunity. The immunity will only bind if the toxin is correctly folded. Although the immunity interaction with the active site could be changed by these mutations, it is not likely that a single amino acid substitution would abolish the interaction between the toxin and immunity altogether. Thus, co-IP in the presence of a cross-linker should be able to confirm that the mutations are not affecting the folding of the toxin. At a minimum, a western blot in the presence of immunity should be performed to show that these toxins can indeed be expressed when sufficient immunity is around.

The toxins which are not detected in the blot (wild type and three mutant versions of Rhs1CT_{DB10}) must be expressed and stable to some degree, since they are toxic to the expressing cells. However

the reviewer makes a good point that the reduced toxicity of the three mutant variants may be due to reduced expression or stability. Therefore we have, as suggested, repeated the western blot to detect the different variants of Rhs1CT_{Db10} in the presence of co-expressed immunity protein. This showed that all the variants are expressed and stable and none show reduced protein levels compared with the wild type toxin. This data is now included in the revised version of the manuscript (Figure 5B).

2. On a similar line, the authors suggest that a basic residue in the immunity is important for neutralization of the Tse6, Tne1 and Rhs NADase toxins respectively. An amino acid substitution to replace the His131 residue in the RhsI protein and showing lack of protection would have strongly strengthened their case.

The reason we did not originally perform this experiment is because the extensive and highly stable nature of the interface between RhsI_{Db10} and Rhs1CT_{Db10} strongly suggests that mutation of any single residue will not be sufficient to prevent interaction between the proteins, occlusion of the active site and, therefore, neutralisation of the toxin.

As noted in the manuscript, 43 residues from RhsI_{Db10} (and 37 residues from Rhs1CT_{Db10}) are involved in interactions between the two proteins. These residues form 16 direct hydrogen bonds and 10 salt bridges between the partners in addition to extensive van der Waals interactions. There are also numerous water-bridged hydrogen bonding interactions that will further contribute to the association of the partners. Thus we considered it highly unlikely that mutation of a single residue in RhsI_{Db10} would abrogate the interaction.

Nevertheless, we have performed the experiment as requested, confirming that mutation of His 131 to alanine does not eliminate function of the immunity protein. This result has been included in the revised manuscript (as new Supplementary Figure 2), together with several sentences in the text explaining how it illustrates the robustness of the effector-immunity interaction, presumably to guarantee protection against the toxin.

3. The finding that some bacteria accumulate Rhs immunity proteins to protect against other toxins is novel and not previously acknowledged. Previous work on other Rhs toxins or their CdiA homologues instead show an accumulation of orphan toxin-immunity modules, suggesting that something is vastly different for the Rhs toxin-immunity pairs identified here. Mentioning this discrepancy could strengthen the importance of the findings.

The Reviewer makes good point with which we agree. We have added a section in the discussion to highlight this new aspect of cross-protection and its distinction from what has been reported previously for Rhs immunity proteins. We have also included a note that putative orphan Rhs immunity proteins in the absence of any Rhs protein have been reported in *Gilliamella* species.

Reviewer #2 (Remarks to the Author):

In this study by Hagan et al., the authors investigate Rhs EI pairs involved in inter-bacterial competition in *S. marcescens* strains. **It is a well-executed study, with a thorough structural investigation of the EI pair and interesting identification of genes encoding orphan immunity proteins.** Although most of the findings in the manuscript have been previously described, the study contributes considerably to the development of knowledge in the field.

We thank the reviewer for the positive comments, and for their thoughts and suggestions below.

There are some general comments and a few minor suggestions for improvement.

1. The authors identified Rhs effectors in two strains that possess NADase toxin domains belonging to different subgroups of this family of proteins. NADase toxins have previously been identified, but it is notable to see that there is a great variety (4 different subfamilies and probably more could be expected) demonstrating their importance for bacterial competition. A phylogenetic analysis of T6SS and non-T6SS NADase toxins would be a beneficial addition to further support the diversity of these toxins and their potential evolutionary relationship.

We appreciate why the reviewer has made this suggestion and the potential benefit of a phylogenetic analysis in understanding the diversity and evolution of related toxins. However, unfortunately, we have concluded that the lack of significant sequence similarity across the different groups of T6SS and non-T6SS NADase toxins means that a phylogenetic analysis across all the groups cannot be performed in a robust and meaningful way. This is why we have relied upon structure-based rather than sequence-based comparisons in the manuscript.

In more detail, we have attempted to compare the NADase toxins Rhs1CT₁₀₃₆, Rhs1CT_{Db10}, TNT and AfNADase with representative sequences from the groups 'Tne1' and 'Tne2', as described in Tang *et al.*, 2018 (10.1074/jbc.RA117.000178). Adopting the methodology used in this previous paper, we used an iterative homology all-against-all search across all sequences mentioned above using jackHMMER, alongside searching for well-curated NADase domains known to be interbacterial toxins, including Bacterial toxin 46 (PF15338) and TNT (PF14021). The iterative all-against-all search using jackHMMER failed to group all sequences together in any of the searches, indicating that the toxin domains cannot be robustly aligned across all of these groups of sequences, due to a lack of sequence similarity. HMMsearch hits to Bacterial toxin 46 (PF15338) corresponded to the Tne1 subgroup, whilst no defined protein domain group corresponded well to the C-terminal toxin domain of Tne2.

As noted in the manuscript, Rhs1CT₁₀₃₆, TNT and AfNADase do show detectable similarity at the sequence level, in addition to clear similarity at the structural level. Our conclusion that Rhs1CT₁₀₃₆ is more closely related to TNT-like NADases than to other T6SS-NADases is further strengthened by the above jackHMMER analysis, since Rhs1CT₁₀₃₆, together with TNT and AfNADase, gives HMMsearch hits to the TNT domain (PF14021). Whilst a few Tne2-like proteins showed some similarity to a small region of the TNT domain according to this analysis, manual inspection suggested that the alignments were unreliable and Tne2-like proteins are not part of this group.

On the other hand, Rhs1CT_{Db10} did not group with any of the other sequences in this analysis, supporting the idea that this NADase belongs to a separate group from all the others, as proposed in the manuscript. We have added a line to the Discussion to note the failure to detect any relation between Rhs1CT_{Db10} and other NADases by this method.

2. The authors used comparative structural analysis to identify conserved residues required for NADase activity and demonstrate it by inhibition assays. They reveal that, as expected, the different families of NADase used non-related immunity proteins, but interestingly they use a shared mechanism of inhibition. The structural section of the manuscript is very comprehensive and well presented in general, but on some occasions, the information is repetitive and could be condensed and combined to improve readability. That is, Figures 1 and 2 could be combined to avoid overlap. The same applies to Figures 4 and 5. The information from Figure 6 is also redundant in Figure 2. Paragraph 234-252 could be restructured, as it repeats information from earlier in the manuscript. The content in Lanes 259-264 is repetitive with the text in the following section. Similarly, the content in lanes 386-388 is repetitive with the content in lanes 401-404.

We have carefully considered the Reviewer's suggestions but have concluded that leaving the Figures separate is required for clarity and logical structure of the manuscript. Combining Figures 1 and 2 results in a Figure that is too large and unwieldy, whilst the Figures as they stand describe two different aspects of the study (Figure 1 is focused on the individual structures of the effector and immunity protein, whilst Figure 2 describes how they interact to form a specific, toxin-neutralising complex). In the original version of the manuscript we did attempt to fit the data in Figures 4 and 5 into the same Figure, but we found that doing so resulted in loss of clarity and legibility. The Reviewer is correct that the structure of the Rhs1CT_{Db10}-Rhs1_{Db10} complex is shown in Figure 2 and Figure 6, albeit in distinct representations showing different features. However Figure 6 forms part of a wider discussion of conservation and divergence of immunity protein mechanism later in the manuscript, and it is not possible to include the content in Figures 2 and 6 in the same Figure whilst retaining clarity. Therefore we prefer to keep the Figures as they are.

We have also carefully checked the sections of text highlighted. A limited amount of repetition was intentional, to help the reader easily understand new observations in the context of those described earlier (for example, saying '*As described above, the structure of the Rhs1CT_{Db10}-Rhs1_{Db10} complex identified an inhibition loop in Rhs1_{Db10} which specifically and effectively blocks the active site of Rhs1CT_{Db10} with the key residue His131*' before moving on to describe the new point that this residue mimics a key interaction of the toxin with its natural substrate and then to describe how this is reminiscent of the mode of inhibition of unrelated immunity proteins), or to allow 'flow' of particular arguments in the discussion. However we have made several edits to the text to ensure that repetition is minimised and it is clearer when new observations are being stated.

3. The study also shows the exchange of CTs between strains, which has been previously demonstrated, and the authors should cite these works. It is especially relevant that the authors acknowledge the pioneering work of Koskiniemi et al., 2014 where they used a relevant setup to show the selection of an evolved population that undergoes recombination of the CT and the immunity pair (<https://www.ncbi.nlm.nih.gov/pmc/articles/PMC3967940/>).

We intended to include this study and its omission was an accidental oversight, now corrected.

4. The authors identified orphan T6SS immunity genes downstream of the study EI pair. T6SS orphan immunity genes have been previously identified in other model organisms, but the magnitude of these orphan immunity proteins is not fully understood and is particularly intriguing. This section of the manuscript could be further expanded with an *in silico* study of the overall distribution of these genes. It would be of great interest to see if the identification of genes encoding orphan immunity proteins in *Serratia* is simply a curiosity, or if it is a frequent phenomenon. It is worth exploring whether these genes produce immunity proteins that protect exclusively against Rhs effectors or if it is a general event that includes protection against other types of effectors as well. Likewise, it is of interest to understand if they appear exclusively in *Serratia* or if they can be found in other genres; and if they are found exclusively in T6SS+ bacteria or any bacteria or T6SS-susceptible microorganisms. Among *Serratia* species, is it common or rare to find T6SS orphan immunity genes? All these questions could be further investigated to provide a more comprehensive understanding of the significance and prevalence of orphan immunity genes in T6SS-mediated inter-bacterial competition.

We agree that all these questions are interesting but are outside the scope of the current study. For example, an *in silico* study of orphan T6SS immunity proteins across bacteria would be a very large undertaking.

We have already referred to literature describing the common occurrence of candidate T6SS orphan immunity proteins in various organisms (lines 55, 422-423, 435-437) indicating to the reader that this is not a *Serratia*-specific curiosity.

Extensive evidence in the field and from our own system indicates that T6SS orphan immunity proteins, like 'in use' immunity proteins, are highly specific for particular effectors and do not provide protection against other types of effectors. For example, it is widely reported that deletion of 'in use' immunity proteins results in a strain susceptible to intoxication by the cognate effector delivered by the parental strain; this would not be the case if orphan immunity genes could provide protection against other effector types. In our model system used in this study, we have previously reported that orphan Tai4-family immunity proteins cannot neutralise non-cognate effectors (DOI: 10.1111/mmi.12028) and observed specific effector-dependent killing of various immunity mutants. Given that immunity proteins function through highly-specific protein-protein interactions with particular effectors, it would not be expected that orphan immunity proteins could protect against different effector types.

5. The immunity proteins described in the manuscript belong to different families and seem specific to their cognate effectors. It would be of interest to test whether they can protect against NADases from other subfamilies. The phylogenetic tree would provide insight into the closest effectors, and specificity could be tested between effector-immunity pairs belonging to the same or different subfamilies.

As noted above, and widely recognised in the literature, protein-protein interactions between effectors and cognate immunity proteins are highly specific. The structures of the distinct effector-immunity complexes shown in this paper indicate that it is extremely unlikely that the immunity proteins could interact with effectors from other subfamilies (residues placed for interactions are missing and/or steric clashes would occur). Figure 6 shows this well, with the overall fold and the nature of residues involved in occluding the active site of the cognate toxin being different between each compared immunity protein.

Moreover, the experiment suggested by the Reviewer has already been done for Rhs1_{Db10} and Rhs1₁₀₃₆. Figure 8E already shows that Rhs1 from Db10 cannot protect against Rhs1CT₁₀₃₆ (since Db10 Δ 2280_{Db10}, which still possesses Rhs1, is intoxicated by Rhs1CT₁₀₃₆ delivered by the engineered attacking strain). Conversely, 2280_{Db10}, which is functionally identical to Rhs1₁₀₃₆, cannot protect against Rhs1CT_{Db10}, since a Δ rhs1_{Db10} mutant is intoxicated by the parental strain delivering Rhs1CT_{Db10} (Alcoforado Diniz and Coulthurst, 2015, the preceding paper for this study). A comment has been added to the relevant results section highlighting these points (and the updated Figure 8b now includes Db10 Δ 2280_{Db10}).

Therefore we do not believe that further testing of cross-protection between subfamilies is needed or would provide any additional information.

6. Rhs effectors have been identified in other delivery systems different from T6SS; the authors should consider discussing these systems further. A relevant example would be described here (<https://journals.plos.org/plosgenetics/article?id=10.1371/journal.pgen.1002217>)

Thank you for the suggestion but given the wide range of functions proposed for polymorphic toxins and other proteins containing Rhs/Wap domains and our desire to keep the current paper as concise and focused as possible, we decided not to discuss these other systems.

7. The authors commented on the fact that the four identified T6SS NADase hydrolysed the oxidised cofactors but not the reduced forms. They could expand this comment including the behaviour of non-T6SS NADases.

The behaviour of AfNADase and TNT has now also been included (it is the same).

Lane 34: The "the" word is not needed before "it delivers", which would read better as "where it

delivers".

Done

Lane 335 and 338. The word 'smaller' could be replaced by 'non-RhS' to refer to PAAR-containing effectors that do not have the core RhS domain.

Done

Lane 464-368. This sentence requires a reference.

Done

Lane 371. Other examples of T6SS nucleases: <https://www.pnas.org/doi/10.1073/pnas.1814181115>; <https://www.nature.com/articles/ismej2016169>

The sentence mentioned by the reviewer reads '*...or produce the dramatic loss in viable target cell recovery observed with some nuclease effectors^{6, 14, 26, 27}...*'. We are not trying to catalogue all T6SS nucleases, but rather provide examples of where T6SS nuclease effectors can cause a dramatic loss in viable target cell recovery during T6SS-mediated inter-bacterial competition. The two examples suggested by the reviewer do not fulfil this criterion, with the former describing only a relatively modest decrease in target cell fluorescence or CFU, and the latter only showing cessation of growth of a population of *E. coli* resulting from artificial expression of a putative nuclease effector.

Lane 405-6. The authors stated that orphan immunity proteins are encoded immediately downstream of intact T6SS EI pairs, but this is not necessarily the case, and arrays of genes encoding immunity proteins can be found even in bacteria without T6SSs. The authors should consider revising this statement and providing more context or examples. Citation number 1 does not seem adequate in this context.

We note firstly that the comment states '*It is common to observe...*' not that they are always located thus. Secondly, we point the reviewer to the end of the same paragraph where we already mention the arrays of immunity proteins and study to which we believe the reviewer is referring ('*...observations by Ross et al. that members of the Bacteroidales possess mobile arrays of genes encoding orphan immunity proteins and that examples of such arrays could provide protection against two effector proteins delivered by the non-canonical Bacteroidales T6SS¹⁰*'). In fact, the reference for this study is already given, with three others, in the place highlighted by the reviewer.

Lane 423-426. This sentence requires a reference.

Done

Lane 427. The importance of RhS effectors is not limited to intra-species competition. The authors should consider generalising this argument.

The comment has been adjusted to make it clear that RhS effectors are important in both contexts.

Lane 440-444. To improve the readability of this statement, the authors should discuss whether these two strains coexist in their natural niche.

This is a general comment about T6SS effectors rather than one related to the two specific strains we have used for the experiment. Additionally, we have no way of knowing whether strains closely related to these particular isolates coexist in a natural niche or not (presumably somewhere they do, given the ubiquity of *Serratia marcescens*). Instead we have edited slightly to try and improve readability.

Figure 2A. The authors should consider whether the upper panel of this figure is necessary or could go to supplementary material.

We feel that it is necessary, and prefer to retain it.

Figure 3. The authors should comment on the fact that the same peaks are at 20 min in panel A and at 30 min in panel B.

The experiments in the two panels are independent and were done separately, some time apart. In the first (panel A), the HPLC gradient was 25 minutes, whilst in the second (panel B) it was 40 minutes, explaining the different elution times for the same compounds. This difference does not affect the outcome or ability to interpret the experiments, since compound standards were run on each occasion and are presented for direct comparison in the top graph in each case. We have clarified this in the Methods section.

Reviewer #3 (Remarks to the Author):

This is a very good research work, however the presentation is not very good. It requires improvement.

This manuscript presents results obtained from combinations of 'Theoretical Biology', 'Molecular Biology', 'Structural Biology' and the 'function' with a main emphasis on 'Heterodimer' 3D-structural result of Rhs1CT_{Db10} - Rhs1_{Db10}. However, it remains unclear until page 8, line 212 that Db10 is the wild type. Moreover, in this manuscript, different results are obtained from different experimental observations and combinations of more than one technique. But at the end, I am not very impressed with the layout, and mode of presentation as:

- i) The results are not well documented.
- ii) It is very difficult to follow the flow of this exciting and wonderful research work.
- iii) Finally, detailing of relevant information is not well covered either.

We thank the reviewer for their comments. We are pleased that they consider our study an exciting and wonderful work and that they appreciate the combination of different experimental approaches we have used. The comments of the reviewer have highlighted a need to clarify that we have made two uses of the word 'model' in the paper: firstly, a 'model' bacterial strain, in the sense of that strain representing a model system; or, secondly, a structural model, whether generated from experimental crystallographic data or *in silico* using AlphaFold. This appears to have created some confusion for the reviewer, for which we apologise and have made appropriate adjustments to the text.

Lists of my concerns are given below –

Comments (C), questions (Q) and criticisms (CS) -

Several abbreviations have been used in this manuscript, so it is better to add the full form and details on their first use (C).

Abstract -

Line (L) 10 – model Db10 or wild type Db10?

3D-structure the heterodimer of wild type strain Db10 – but not clear here.

Was the subsequent comparative analysis of Db10 and SJC1036 done using the model of SJC1036?

We have edited the abstract to clarify that Db10 is a model bacterial strain. All the structural analyses were done on wild type proteins, i.e. proteins with the sequence encoded by the wild type strains in question. The abstract does not state that comparative analyses was performed between Db10 and SJC1036. A variety of comparative structural analyses involving and between the different NADase effector domains were performed in the study and we feel that the sentence provided in the manuscript is an accurate summary of the overall conclusion from these analyses.

Introduction -

Page 3

Line (L) 24) – In my opinion before the beginning of T6SS functions, it would be better to add a few words about the effectors Hcp, PAAR etc. – which would make it easier for the readers to follow the text (C) mentioned on L34.

Experience has told us that it is generally clearer for non-specialists to be first introduced to the overall function of the T6SS (*'a large, contractile nanomachine used to deliver toxic effector proteins into neighbouring cells'*) before describing individual structures and components of the system. Therefore we prefer to retain the current order.

Page 4

L 66 – 67 – 'The bacterium *S. marcescens* is present in diverse environmental conditions and is responsible in hospital-acquired infections' –

i) What are these and how severe are these infections?

Details of the nature and severity of the highly varied hospital acquired infections caused by *Serratia marcescens* are not relevant to the current work. The reader may consult the cited reference for more information if they are interested.

L 69 – 'at least ten effector proteins, including two Rhs proteins' – one is Rhs1, what is the other?

Rhs2

L-81 – 'orphan immunity protein gene – add the details of this gene here (C).

The details are provided in the Results section when the gene is identified, introduced and characterised. This is simply a summary sentence of the key conclusion.

Page 5

(C) – At the beginning the details of residues, the range of Rhs1CT_{Db10} - Rhs1L_{Db10} complex should be added. The main reason is that, on this page, lots have been discussed about the structure, however there is no mention about the Mw of this complex and what was the span of residues that it constitute.

We refer the reviewer to Figure 1, where the amino acid sequence, residue numbers and secondary structure elements of both proteins in the structure of the complex are given.

The Supplementary Tables contain precise details of the nucleotide sequence and amino acids encoded in each construct, and the theoretical MW of the complex and individual proteins is given in the Methods.

Nevertheless, we have added the range of residues which comprise the CT domain and the molecular weight of the complex to the text on this page as requested.

L 101 - 102 – Is there any specific nature of the residues, e. g. electro +ve/-ve, acidic etc. which are on top of the putative active site? (Q)

In response to the reviewer's question, there are only polar residues with no prevalence of a particular charge in the area extending out from the putative active site and onto the surface of the protein.

Page 6

L124 -126 – D129 – N132 loop of Rhs1_{Db10} resulted to active site blockage of Rhs1CT_{Db10} – hence it can be classified as closed or inactive structure – are there any comments on the opening and closing of the active site?

The nature of the protein-protein complex structure suggests to us that only limited conformational changes might be implicated in activity. However, we are unable to determine if the putative active site is in an open or closed conformation, or indeed if localised conformational alterations might be relevant to function. The inhibition by the immunity protein involves a high affinity interaction between two proteins. It would not be possible to determine if the active site is open or closed unless we were able to determine the structures of both states of the enzyme, or at the very least a structure showing a large opening of the cleft. We judge it sensible to avoid undue speculation on this point.

For the Reviewer's interest, we did compare the structure of Rhs1CT_{Db10} in complex with Rhs1_{Db10} with the AlphaFold prediction for Rhs1CT_{Db10} alone. There was nothing to suggest that in the absence of immunity the active site of Rhs1CT_{Db10} would undergo conformational change.

L 137 – 139 – What is the sequence homology?

The proteins do not align in a robust or convincing way at the sequence level. Sequence identities are less than 15% even over the more conserved region. Therefore, we have chosen to only include numbers quantifying the structural alignment.

L 141 – PDB code of NADase is missing. (In the PDB there are three structures NADase is 6YGE the other two. 6YGF and 6YGG are complex with various ligands.

The three *Af* NADase structures are so similar that Dali does not distinguish between them when comparing this protein with Rhs1CT_{Db10} and only outputs the scores for one out of three accession codes (6YGF). Therefore, we have now included the 6YGF code, since this is the structure selected to be reported by Dali as the hit.

Whilst checking this point, we noticed that the manuscript contained r.m.s.d. values generated at different points during the study and therefore not calculated in exactly the same way at the same time. We have repeated these calculations in a uniform manner, resulting in small changes to several numerical values but with no impact on the interpretation or conclusions.

Page 7

L 175 and 403 – 'Presence of a Cationic interaction/intermediate or a transition state - - which support breakage of C – N bond and the formation of C - OH bond' – any experimental evidence in support of this statement? (Q)

In response to the reviewer's question, we have no experimental evidence to support or refute this aspect of the proposed mechanism. We would hope that the editor will appreciate the challenge that always accompanies the discussion of an enzyme mechanism given the difficulty of being able to access structural data relating to the key stage, i.e. the transition state. Our discussion here is simply a sensible explanation of a plausible mechanism based on established chemical principles.

Page 8

L 199 - 201 – Either you must propose what are the possible potential contributors to validate the enzyme mechanism or have to remove the entire sentence – (C).

We have done so, in the section above. The highlighted sentence is simply intended to be a concluding comment to the section. We have adjusted the wording to make this more clear.

Page 9

L236 – 238 - Rhs1_{Db10} fold – Is it really a new fold? Could you please check again?

- To me it looks to be a very common β -sandwich, β -trefoil and other common β -sheet dominated protein fold.

- This is only my comments, because I cannot check myself without the coordinate.

The coordinates resulting from our study have been made available via the PDB in advance of our manuscript submission (noted in the manuscript, reporting summary and PDB validation report) and the reviewer could have accessed the model to further investigate the idea of a new fold.

The issue here is one of semantics. Our view is that if the polypeptide shows an arrangement of the elements of secondary structure along the polypeptide that has not been previously observed then the fold of that polypeptide is new. Indeed, the structure of the immunity protein is dominated by β -sheets but it would not be possible to align the structure with other β -sandwich proteins in a manner that also aligns the amino acid sequences.

L 240 – ‘As observed H131 is the key residue to block the active site’ – was there any attempt to mutate H131?

- If so, what was the observation?

- My own and readers main interest would be to know whether H131 is the key in opening and closing of the active site. In my knowledge, I know few structures where a single residue was key in opening and closing the active site of the structure.

- In others, the key was the residue along with other interactions.

We have included new data on mutation of H131 in response to Reviewer 1.

We have simply noted that H131 in the immunity protein plays a key role in blocking the active site of the toxin. We have not made any claims or comments about ‘opening and closing’ of the active site of the toxin (see also above), and this active site is in a different protein from H131 in any case.

Page 10

Fig 7A – structural model of Rhs1₁₀₃₆ – based on which structure the model built? (Q)

- What are the major differences?

As stated in the text (lines 271-272), the structural model was generated using AlphaFold2.

Therefore it was not built using a particular structure and the question about ‘major differences’ is not applicable.

- What is the sequence homology of Rhs1_{Db10} and Rhs1₁₀₃₆? (Q)

There is no detectable homology. The proteins are unrelated.

Fig 7B - Superposition of Rhs1CT_{Db10} and Rhs1CT₁₀₃₆ – it looks like Rhs1CT₁₀₃₆ has a deletion peptide – if so, how long is that peptide?

We are not clear what the Reviewer is referring to here. Rhs1CT₁₀₃₆ does not contain any deletions and is 13 amino acids longer than Rhs1CT_{Db10}.

L 290 and Fig 8B – ‘Comparison of Db10 and SJC1036 genome around *rhs1* - revealed two genes ---: In Db10 9i) Rhs1_{Db10} (ii) 2280 whereas in SJC1036 Rhs1₁₀₃₆ is missing but similar to 2280 another gene is present’ – is that a homologue of Rhs1_{Db10}? It is not at all clear from the text on line 290 - 291.

We have adjusted the text in lines 290-291 to improve clarity. As noted below, we have also adjusted Figure 8A by the introduction of colour coding to indicate homologous genes, to assist

readers who are less familiar with synteny plots. This should also more clearly indicate the genes and domains that are present and conserved, or missing, between the the genetic loci of the two strains.

Page 33 – NADase PDB is not 6YGG rather 6YGE is complex of NADase with various ligands.

PDB 6YGG is *Af* NADase complexed with a substrate analogue, whilst PDB 6YGE is *Af* NADase. The alignment in Figure 7 (page 33 in the original version) was made using PDB 6YGG and so the legend is correct. The presence of a ligand does not detectably alter the structure of *Af*NADase so the alignment would be unchanged whichever was used.

Page 11

L 301 -327 – (C)

- Well planned work, however the presentation of this section is not up to the standard; in my opinion this could have been presented much better.
- To me, this section is overwhelmed, congested, and very difficult to follow.
- I would add a small table and less text, to make this section very attractive and easy for the researcher/reader to follow.

We have revised Figure 8 to be as clear as possible and better aid the reader in understanding the genetic manipulations and experiments that were performed. In panel A, we have added colour coding to indicate homologous genes, using the same colour scheme as in panel B. Panel B has been redrawn to depict gene rather than protein organisation and therefore match panel A. It also now shows the structure of the *rhs1* locus in all the strains derived from *S. marcescens* Db10, namely the wild type and all the engineered and mutant derivatives (but it does not include SJC1036 and *S. ficaria* as the corresponding information is already present in panel A).

We have taken on board the Reviewer's opinion that this section may have been hard to follow for some readers. We do not believe that introducing a table in place of text would be appropriate, but instead we have revised Figure 8 to provide clearer visual representation of the genetic loci and genetic constructions described in this section. We have also made adjustments to reduce the density and improve the clarity of the text in this section.

Page 12

L 357 – 'The existence of four distinct Tne groups' – Tne1 and Tne2 sub-groups have already been well established in T6SS, but not yet for Tne3 and Tne4.

- At this stage, I am reluctant to accept the existence of four distinct subgroups of Tne, instead there may be two other new sub-groups 3 & 4.

We are not clear exactly what point the Reviewer is trying to make here. We note that we already say that 'Rhs1CT_{Db10} and Rhs1CT₁₀₃₆ may represent new sub-groups, Tne3 and Tne4'

However please see the response to Reviewer 2, point 1, where we explain how the use of iterative all-against-all searching using jackHMMER provides further support for our proposal that there are at least four distinct sub-groups of Tne, with this study identifying the founding members of the Tne3 and Tne4 subgroups.

Page 17 – JCSG screen reference is missing.

The manufacturer of this commercial product is noted. We have corrected the name to JCSG-plus™.

Page 18 –

L 541 – Asn24 and Leu155 – is not quite right – I think those are Asn1342 and Leu1473?

Yes, the Reviewer is correct, thank you for alerting us to this error. We have corrected the numbering.

Page 19 – No Fig 5C

Removed.

L586 – What is MeCN?

Acetonitrile. We have replaced 'MeCN' with 'acetonitrile' in both places.

Page 21 – Reference 22 and 23– incomplete.

Fixed (reference numbers are now 22 and 24)

My decision is – Acceptance and publication of this manuscript subject to clarification and appropriate modifications listed above.

REVIEWERS' COMMENTS

Reviewer #2 (Remarks to the Author):

The revision carried out by the authors is appropriate, although limited. I have no further comments to add.